# Natural epigenetic polymorphisms lead to intraspecific variation in Arabidopsis gene imprinting

**Daniela Pignatta[1], Robert M Erdmann[1,2], Elias Scheer[1], Colette L Picard[1,3], George W Bell[1], Mary Gehring[1,2,3]***

[1]Whitehead Institute for Biomedical Research, Cambridge, United States; [2]Department of Biology, Massachusetts Institute of Technology, Cambridge, United States; [3]Computational and Systems Biology Graduate Program, Massachusetts Institute of Technology, Cambridge, United States

**Abstract** Imprinted gene expression occurs during seed development in plants and is associated with differential DNA methylation of parental alleles, particularly at proximal transposable elements (TEs). Imprinting variability could contribute to observed parent-of-origin effects on seed development. We investigated intraspecific variation in imprinting, coupled with analysis of DNA methylation and small RNAs, among three Arabidopsis strains with diverse seed phenotypes. The majority of imprinted genes were parentally biased in the same manner among all strains. However, we identified several examples of allele-specific imprinting correlated with intraspecific epigenetic variation at a TE. We successfully predicted imprinting in additional strains based on methylation variability. We conclude that there is standing variation in imprinting even in recently diverged genotypes due to intraspecific epiallelic variation. Our data demonstrate that epiallelic variation and genomic imprinting intersect to produce novel gene expression patterns in seeds.

*For correspondence:
mgehring@wi.mit.edu

**Competing interests:** The authors declare that no competing interests exist.

**Reviewing editor**: Detlef Weigel, Max Planck Institute for Developmental Biology, Germany

## Introduction

Diploid sexually reproducing organisms inherit an allele of each gene from both parents, which masks deleterious effects of recessive mutations. However, a subset of genes in flowering plants and mammals are subject to imprinting, whereby genes are expressed predominantly from one allele in a parent-of-origin dependent manner, such that traits controlled by these genes reflect the genotype of only one parent. Imprinted gene expression affects fetal growth regulation and postnatal behavior in mammals and the formation of viable seeds and the inhibition of interspecies hybridization in plants (*Tycko and Morison, 2002*; *Jiang and Köhler, 2012*; *Kradolfer et al., 2013*). Imprinting is primarily restricted to the endosperm in plants, the triploid tissue that develops alongside the embryo and is necessary for normal embryo patterning and growth. Genome-wide surveys for imprinted expression have identified several dozen to hundreds of imprinted genes in rice, maize, Arabidopsis, mice, mules, and hinnies (*Gehring et al., 2011*; *Hsieh et al., 2011*; *Luo et al., 2011*; *Wolff et al., 2011*; *Zhang et al., 2011*; *DeVeale et al., 2012*; *Waters et al., 2012*; *Xin et al., 2013*; *Wang et al., 2013b*). Differential DNA methylation and histone modifications between maternally-inherited and paternally-inherited alleles are important regulators of imprinted gene expression (*Köhler et al., 2012*). Endosperm DNA is maternally hypomethylated at thousands of discrete loci (*Gehring et al., 2009*; *Hsieh et al., 2009*; *Ibarra et al., 2012*) in a process dependent on the 5-methylcytosine DNA glycosylase DME (*Gehring et al., 2006, 2009*; *Hsieh et al., 2009*; *Ibarra et al., 2012*). Maintenance and de novo methylation pathways also appear to be compromised in the central cell and during early endosperm development (*Jullien et al., 2012*; *Belmonte et al., 2013*; *Vu et al., 2013*), which might further contribute to loss of methylation from the maternally inherited genome. Endosperm

**eLife digest** When animals or plants reproduce sexually, the DNA in a sperm or pollen is combined with that in an egg cell to generate an offspring that inherits two copies of each gene, one from each parent. For a very small number of genes, the copy from one of the parents is consistently turned off. This process—called imprinting—means that the same gene can have different effects depending on if it is inherited from the mother or the father. In plants, imprinting is vital for the production of seeds and typically occurs in the endosperm: the tissue within a seed that provides nourishment to the plant embryo.

One way genes can be imprinted is by adding small chemical marks—called methyl groups—on to the DNA that makes up the gene or nearby sequences. These marks can either switch on, or switch off, the expression of the gene. DNA methylation also immobilises stretches of DNA called transposable elements, stopping them from moving from one location to another in the genome. These stretches of DNA are identified and targeted for methylation by small molecules of RNA that match their DNA sequences.

Genes that are imprinted in the endosperm of the model plant Arabidopsis are often associated with transposable elements, which can be methylated differently in the naturally occurring varieties, or strains, of Arabidopsis. However it is unclear how many genes are differently imprinted between these different strains.

Pignatta et al. looked for differences in gene imprinting, DNA methylation and small RNA production in the seeds, embryos and endosperm tissue from three strains of Arabidopsis. They also examined seeds from crosses between these three strains.

While most genes had the same imprinting pattern in all strains and crosses examined, 12 genes were imprinted differently depending on whether they were inherited from the male or female of a given strain. For example, for some genes the copy inherited from the male parent is always turned off, unless it is inherited via the pollen of one specific Arabidopsis strain. Half of this variation could be explained by a transposable element near to each gene that was methylated differently among the strains.

By comparing the differentially methylated regions in the genomes of 140 Arabidopsis strains, Pignatta et al. found that differences in methylation may affect 11% of imprinted genes—and went on to confirm variable imprinting in some Arabidopsis strains based on the presence or absence of DNA methylation.

Future work is needed to understand how variation in gene imprinting might affect the traits of hybrid seeds, and how it might affect the evolution of new traits in hybrid plants.

DMRs (differentially methylated regions) are enriched for TE sequences, although not all imprinted gene are associated with a neighboring TE (*Gehring et al., 2009*).

TE methylation dynamics during reproduction appear to be an important driver of imprinted gene expression, yet the epigenetic modification of TEs and their presence or absence in genomes can be variable on short evolutionary timescales. Although very few TEs are presently active in *Arabidopsis thaliana*, most differences between genomic sequences of Arabidopsis strains are due to variation in TEs (*Cao et al., 2011*). Many euchromatic TEs and related sequences are targeted for RNA-directed DNA methylation (RdDM), an active process through which 24-nt small RNAs derived from longer non-coding RNA transcripts direct DNA methyltransferases to cognate sequences (*Law and Jacobsen, 2010*). RdDM is important for maintaining transcriptional silencing of TEs. Gene expression is negatively correlated with the proximity of TEs targeted by small RNAs and methylated TEs are under stronger purifying selection when they are near genes (*Hollister and Gaut, 2009*; *Wang et al., 2013a*). TE methylation is quite stable, although loss of methylation at TEs can occur spontaneously at very low frequency (*Becker et al., 2011*; *Schmitz et al., 2011*). However, a quarter of TEs in Arabidopsis are not methylated (*Ahmed et al., 2011*) and only 68% are associated with small RNAs (*Hollister et al., 2011*). Thus the epigenetic modification status of TEs can be variable within the species, between different classes of elements, or even among elements of the same family.

The potential role of TEs in establishing or maintaining imprinted expression coupled with the evolutionary forces that select for parent-of-origin specific expression suggest that substantial intraspecific

variation in imprinting could exist. Indeed, the first imprinted gene described, the maize *R* gene (*Kermicle, 1970*), is an example of allele-specific imprinting; only alleles that have a *Doppia* TE inserted in the promoter are imprinted (*Kermicle, 1978*; *Walker, 1998*; *Alleman and Doctor, 2000*). Once imprinted gene expression arises, the kinship or parental conflict theory of imprinting (*Haig, 2013*) posits that it could be evolutionarily selected because asymmetrically related kin (e.g., half-siblings that have the same mother but different fathers) compete for maternal resources. Thus, maternally and paternally inherited alleles of genes that influence maternal resource transfer to offspring have different optima for total gene expression levels. Plants adopt a range of different strategies with regards to maternal resource transfer to offspring—producing a few large seeds, or many small seeds. Intraspecific variation in this trait could potentially be linked to differences in the set of genes subject to imprinting in each strain. To systematically evaluate whether gene imprinting varies on short evolutionary time scales and to further understand the role of genetic and epigenetic polymorphisms in this process, we have investigated the conservation and variability of imprinting, DNA methylation, and small RNA production in reciprocal crosses among three strains of Arabidopsis. Arabidopsis is an ideal system in which to ask these questions because of the availability of genotyped and epigenotyped strains that have diverged for only a few thousand years.

Here we discovered 12 examples of allele-specific imprinting, about half of which were associated with endosperm demethylation of a TE that was variably methylated within the strains we examined. We further evaluated intraspecific methylation variability at regions targeted for CG DNA demethylation during female reproductive development for 140 strains where vegetative methylation patterns are known (*Schmitz et al., 2013*). Approximately 11% of imprinted genes are associated with an endosperm DMR that is variably methylated among strains. From this analysis we predicted and experimentally validated allele-specific imprinting in additional strains for two genes. The ability to predict imprinting status based on strain-to-strain variation in vegetative methylation patterns suggests that these genes are strong candidates for allelic variation in imprinting due to epigenetic differences at TEs, and thus act as epialleles. Our data demonstrate that epiallelic variation and genomic imprinting intersect to produce novel gene expression patterns in seeds. Thus, naturally occurring epialleles could have the strongest phenotypic effect during the reproductive phase of plant development, when patterns of methylation are altered.

## Results

We analyzed genome-wide DNA methylation, small RNA, and gene expression patterns in whole seeds, embryo, and endosperm of three different *Arabidopsis thaliana* strains and in reciprocal crosses among them to investigate imprinting variation within the species and to assess the role of epigenetic polymorphisms in this process. We focused on the commonly used strains Col and Ler and the more polymorphic strain Cvi (*Nordborg et al., 2005*). Cvi produces larger seeds than Col or Ler (*Figure 1—figure supplement 1*) due to increased and prolonged endosperm and integument growth (*Alonso-Blanco et al., 1999*), an effect that is even more pronounced when Cvi is pollinated by Col or Ler (*Figure 1—figure supplement 1*). In our experimental design the Col, Ler and Cvi alleles for each gene were represented twice as the maternally or paternally inherited allele, allowing us to draw conclusions about the activity of an allele independent of the genotype of the other parent in the cross.

### Allele-specific expression analysis identifies genes imprinted among all strains

We first identified genes with consistent parentally biased expression in embryo and endosperm using mRNA-seq data from six different crosses representing three sets of reciprocals: Col-Ler, Col-Cvi, and Ler-Cvi (*Figure 1*, *Figure 1—source data 1*). Imprinting could be evaluated for 16,646 loci in at least one set of reciprocal crosses and 8088 loci in all three sets of reciprocal crosses based on SNPs and sequencing depth. As previously, we implemented a series of filters to define imprinted genes (*Gehring et al., 2011*) ('Materials and methods') except that we added an additional filtering step to require endosperm maternally biased genes to have at least 85% maternal reads in each direction of the cross and paternally biased genes to have at least 50% paternal reads in each direction of the cross (the maternal and paternal cutoffs in the embryo were 70%) (*Figure 1*). In the endosperm between 122 and 145 maternally expressed imprinted genes (MEGs) were identified for each pair of reciprocal crosses (Col-Ler, Col-Cvi, or Ler-Cvi) along with between 43 and 52 paternally expressed imprinted genes (PEGs) for a total of 285 possible MEGs (including 5 TEs) and 103 PEGs in the union of all crosses (*Figure 1B*,

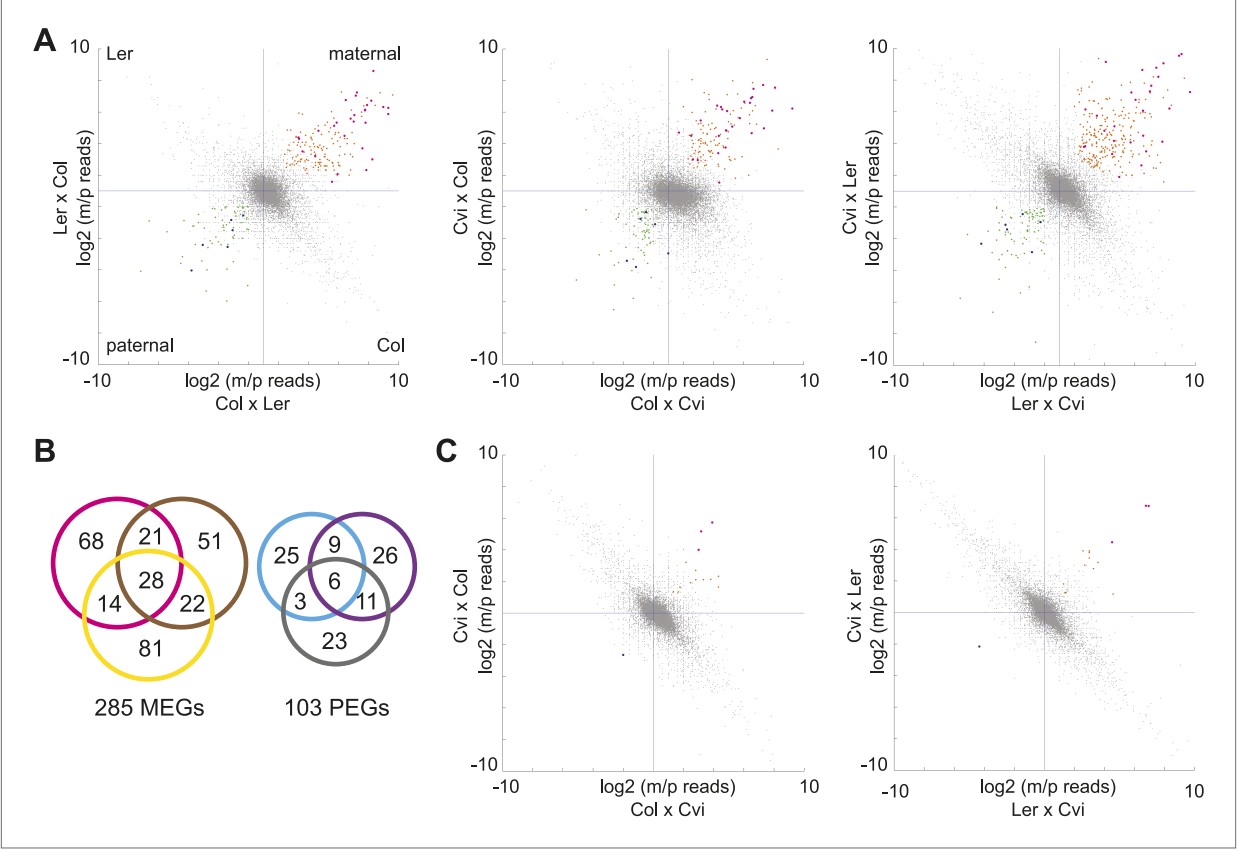

**Figure 1**. mRNA-seq identifies genes with biased expression. (**A**) Proportion of maternal (m) and paternal (p) reads for all three sets of reciprocal crosses in the endosperm. One replicate of each reciprocal cross is shown. Biases represented by each quadrant are depicted for Col-Ler endosperm crosses but apply to all graphs. Orange and pink dots represent MEGs (pink dots are MEGs in all three sets of reciprocal crosses), blue and green dots represent PEGs (blue dots are PEGs in all three sets of reciprocal crosses). Crosshairs indicate the expected log ratio for genes that lack biased expression. (**B**) Overlap of MEGs and PEGs in the endosperm among three sets of reciprocal crosses. Pink and blue circles: Col-Ler; brown and purple circles: Col-Cvi; yellow and gray circles: Ler-Cvi. (**C**) Proportion of maternal (m) and paternal (p) reads for Col-Cvi and Cvi-Ler reciprocal crosses in the embryo. Colored dots as in part **A**. *Figure 1—figure supplement 1* shows seeds used in the experiment. *Figure 1—figure supplement 2* shows validation of an imprinted gene. *Figure 1—figure supplement 3* examines maternal:paternal ratios of imprinted genes identified in one set of crosses in the other two sets of reciprocal crosses. *Figure 1—figure supplement 4* examines overall expression levels of imprinted genes at other stages of development. Information on mRNA-seq library metrics is in *Figure 1—source data 1* and allele-specific expression information for all genes in endosperm and embryo is in *Figure 1—source data 2* and *Figure 1—source data 3*, respectively. *Figure 1—source data 4* shows the overlap among imprinted genes identified in this study and those identified in previous efforts and *Figure 1—source data 5* includes independent validation of imprinted genes.

The following source data and figure supplements are available for figure 1:

**Source data 1**. mRNA-seq libraries generated in this study.

**Source data 2**. Endosperm imprinting data for all genes.

**Source data 3**. Embryo imprinting data for all genes.

**Source data 4**. Overlap among published imprinted gene lists.

**Source data 5**. Validation of imprinted genes.

**Figure supplement 1**. Seed development in the crosses used in this study.

**Figure supplement 2**. Validation of AT4G00750 allele-specific imprinting by RT-PCR and CAPs digestion.

*Figure 1. Continued on next page*

*Figure 1. Continued*

**Figure supplement 3**. Consistency of imprinting among different sets of reciprocal crosses.

**Figure supplement 4**. Imprinted genes are expressed at multiple stages of development.

*Figure 1—source data 2*). Many of these genes have previously been identified as imprinted genes (*Figure 1—source data 4*). Consistent with previous results (*Gehring et al., 2011*; *Hsieh et al., 2011*), very few potential imprinted genes were detected in the embryo (*Figure 1C*, *Figure 1—source data 3*). Imprinting calls based on whole-genome mRNA-seq were validated by sequencing or performing CAPs digestion on RT-PCR amplicons of 29 genes from independently isolated embryo and endosperm RNA samples (*Figure 1—figure supplement 2*, *Figure 1—source data 5*); results were mostly consistent with the mRNA-seq data.

We concluded that most genes that show strong evidence for imprinting in one cross have evidence for the same parental bias in other crosses (*Figure 1—figure supplement 3*). The intersection of the endosperm datasets revealed 28 MEGs and 6 PEGs in common among all three pairs of reciprocal crosses (*Figure 1B*). An additional 53 MEGs and 23 PEGs were identified in two of three sets of reciprocal crosses. Most MEGs and PEGs that were identified in only one set of reciprocal crosses lacked sufficient data to assess imprinting in the other crosses, due to an absence of SNPs or because of low read counts, rather than because they were clearly not parentally biased. For example, of the 73 Col-Cvi MEGs that were not among the Col-Ler MEGs, 62 lacked sufficient allele-specific data to be assessed for imprinting in Col-Ler. Of the remaining 11 genes, 8 showed strong evidence for maternal bias but did not meet all criteria for imprinting (usually failing to meet the requirement for 85% maternal reads in both directions of the cross).

We also examined the expression of imprinted genes at other stages of the plant life cycle (*Figure 1—figure supplement 4*). Analysis of published microarray expression data (*Toufighi et al., 2005*) showed that in Arabidopsis most imprinted genes are expressed at other stages of plant development (*Figure 1—figure supplement 4*). Within seeds, imprinted genes are most commonly expressed in chalazal endosperm (*Figure 1—figure supplement 4*), consistent with previous findings from individual loci (*Ingouff et al., 2005*). Several PEGs and MEGs are most highly expressed in mature pollen (*Figure 1—figure supplement 4*), probably reflecting expression in the pollen vegetative nucleus, which also undergoes active DNA demethylation (*Schoft et al., 2011*; *Calarco et al., 2012*; *Ibarra et al., 2012*).

Consistent with our previous study (*Gehring et al., 2011*), compared to all genes that could be assessed for imprinting, the PEGs identified in any of the three sets of reciprocal crosses (n = 103) were enriched for genes encoding proteins with a SRA-YDG domain (65.6-fold; p value=3.3E−6), genes involved in the biological processes regulation of RNA metabolic processes (4.1-fold; p value=4.7E−4), DNA-dependent regulation of transcription (4.1-fold, p value=8.5E−4), and DNA binding proteins (2.6-fold; p=2.5E−4). Overall, PEGs consisted of many genes known or predicted to be involved in transcription and epigenome regulation. Maize PEGs are also enriched for chromatin modifiers (*Waters et al., 2013*). The Arabidopsis MEGs were not enriched for any particular class of genes except for a slight enrichment for transcription factor activity (2.1-fold; p=0.035), particularly of the MYB and homeodomain types.

## A small number of genes exhibit allele-specific imprinting

We identified 9 PEGs and 3 MEGs that exhibited allele-specific imprinting (*Figure 2—source data 1*). Our method to identify imprinted genes explicitly relies on agreement between reciprocal crosses ('Materials and methods'). However, genes that exhibit allele-specific imprinting will only be parentally biased when a particular strain is the male or female parent; for example a gene could be a PEG in all crosses except when Cvi is the male parent (*Figure 2*). From the ratio of maternal/paternal mRNA-seq reads in each cross we identified those loci that were potentially biased in one direction of the cross but not the other ('Materials and methods'). Because each allele was included twice as the maternal or paternal parent in our experimental design, we were able to identify loci that consistently showed maternal or paternal bias when a particular strain was the male or female parent (*Figure 2*). These lists were then compared to the union of all MEGs and PEGs (*Figure 1*) to identify genes that are imprinted

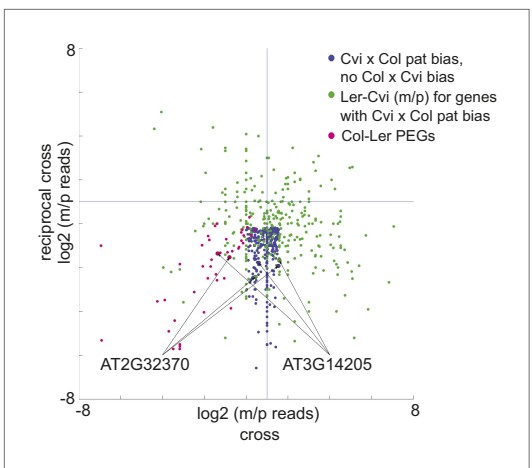

**Figure 2**. A subset of genes is only imprinted when a certain strain is the male or female parent. Process for identifying allele-specific imprinted genes that are PEGs except when Cvi is the male parent. Genes that are paternally biased in Cvi x Col but not Col x Cvi (blue dots) were identified. These genes were overlapped with the Ler-Cvi maternal/paternal log ratios for the same genes (green dots) to generate a list of candidate loci that are not PEGs when Cvi is the male parent. Intersection with Col-Ler PEGs (pink dots) identifies strain-specific imprinted genes that are PEGs except when Cvi is the male parent, including AT2G32370 and AT3G14205. All candidate allele-specific imprinted genes are in **Figure 2—source data 1**. m, maternal; p, paternal.

The following source data is available for figure 2:

**Source data 1**. Candidate allele-specific imprinted genes.

in one set of reciprocal crosses, but only in one direction in the other two sets of reciprocal crosses (**Figure 2**). We performed CAPs analysis or sequenced RT-PCR amplicons for 7 of the 12 allele-specific imprinted genes and further confirmed that they exhibited allele-specific imprinting with few exceptions (**Figure 1—figure supplement 2**, **Figure 1—source data 5**, **Figure 2—source data 1**).

## Embryo-endosperm DMRs are largely distinct from strain DMRs

To explore the potential relationship between DNA methylation and conserved and allele-specific imprinting, libraries for single base resolution DNA methylation profiling by bisulfite sequencing (BS-seq) were prepared from embryo and endosperm DNA (**Figure 3—source data 1**). All BS-seq libraries had a high cytosine to thymine conversion rate (at least 99.70%), indicating efficient bisulfite treatment (**Figure 3—source data 1**). Methylation profiles were generated from non-redundant uniquely mapping reads ('Material and methods'). Endosperm DNA was consistently less methylated compared to embryo DNA in all sequence contexts (**Figure 3—source data 1**), as shown previously (**Gehring et al., 2009**; **Hsieh et al., 2009**; **Ibarra et al., 2012**). Embryo tissue from crosses between strains primarily displayed additive total DNA methylation (**Figure 3—source data 1**). In the endosperm, total methylation was more closely aligned with the methylation level of the female parent, consistent with the 2:1 ratio of maternal to paternal genomic DNA in the endosperm (**Figure 3—source data 1**). Interestingly, CHH methylation was not substantially reduced on a global scale in Cvi x Col endosperm (3.0%) compared to the embryo (3.1%) (**Figure 3—source data 1**).

Our analysis revealed an unusual methylation profile in Cvi. CG methylation in Cvi embryos (16.3%) was lower than in Col and Ler embryos (28.1% and 22.4%, respectively) (**Figure 3—source data 1**). We further investigated the nature of CG hypomethylation in Cvi to determine whether it was specific to certain classes of sequences or whether CG methylation was uniformly reduced across the genome. Of the three strains, Cvi had the lowest median CG methylation levels in both genes and TEs in embryos (**Figure 3**), but was largely unaffected in those sequences in either the CHG or CHH contexts (**Figure 3C**, **Figure 3—figure supplement 1**). Analysis of Cvi CG methylation data from leaves (**Schmitz et al., 2013**) confirmed that hypomethylation was not specific to embryo and endosperm but is a general property of this strain (**Figure 3—figure supplement 1**). Loss of CG methylation was most pronounced in gene bodies (**Figure 3B,C**, **Figure 3—figure supplement 1**), where it was 50% lower in Cvi compared to either Ler or Col (**Figure 3—source data 2**). CG methylation in TEs was 14% lower in Cvi than in Col, but at the same level as in Ler (**Figure 3—source data 2**). Reduced gene body methylation in Cvi is unlikely to be a technical artifact of mapping biases: gene bodies have fewer SNPs and indels than TEs and thus mapping efficiency is better to genes than to TEs. All of our locus-specific bisulfite sequencing confirmed the whole-genome BS data. In Arabidopsis, gene body methylation is primarily in the CG context and is maintained after DNA replication by the maintenance methyltransferase MET1 (**Lister et al., 2008**). Therefore, non-CG methylation pathways seem to operate normally in Cvi, but maintenance methylation appears to be compromised.

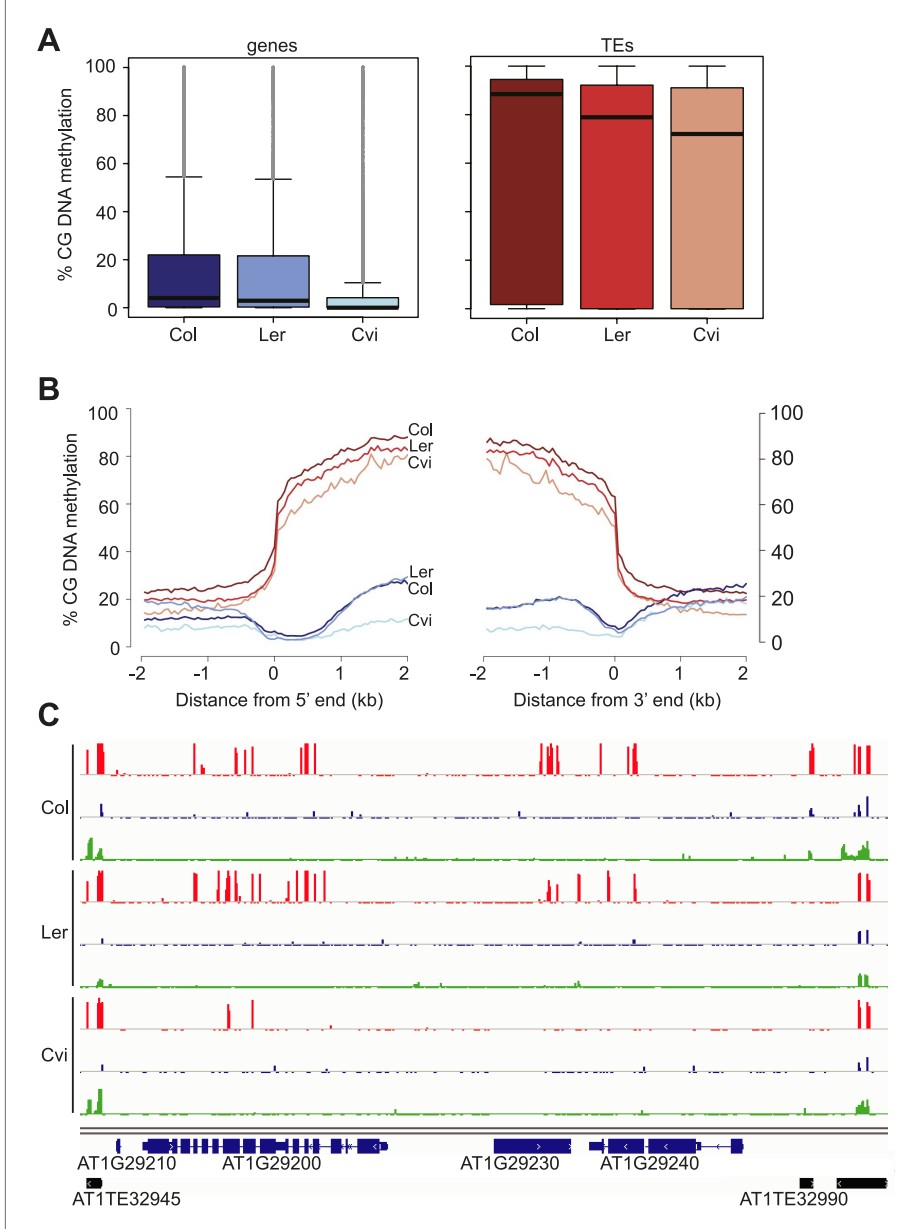

**Figure 3**. Cvi is hypomethylated in CG contexts. (**A**) Box and whiskers plots of CG DNA methylation levels of genes and TEs in Col, Ler, and Cvi embryos. Line: median; gray dots: outliers. (**B**) Average CG DNA methylation profiles of genes (blue colors) and TEs (orange colors) in Col, Ler, and Cvi embryos. Relative to Col, mean Cvi methylation level was decreased by 56% in genes (p=0.00, Tukey's HSD test) and by 14% in TEs (p=0.00, Tukey's HSD test). (**C**) DNA methylation in Col, Ler and Cvi embryos at a representative genomic region that includes genes and TEs. CG (red), CHG (blue) and CHH (green) methylation. Tick marks below the line indicate cytosines for which data was present but no methylation was detected. *Figure 3—figure supplement 1* contains additonal analyses, *Figure 3—source data 1* has statistics on BS-libraries and *Figure 3—source data 2* shows the complete statistical analysis of methylation in Cvi compared to other strains.

The following source data and a figure supplement are available for figure 3:

**Source data 1**. BS-Seq libraries generated in this study.

**Source data 2**. Statistical analysis of strain differential methylation in genes and TEs.

*Figure 3. Continued on next page*

*Figure 3. Continued*

**Figure supplement 1**. Cvi is hypomethylated in CG contexts regardless of tissue type but is not as hypomethylated in CHG and CHH contexts.

Genomic regions that are subject to active DNA demethylation during endosperm development in at least one strain but are variably methylated among other strains could be important contributors to an epigenetic mechanism for allele-specific imprinting. To identify embryo-endosperm DMRs and strain DMRs, we calculated weighted methylation levels (*Schultz et al., 2012*) in overlapping 300 nucleotide windows across the entire genome, requiring each cytosine to have at least five reads coverage to be included in the analysis ('Materials and methods'). We then ran pairwise comparisons between all windows with sufficient coverage for embryo libraries and their matching endosperm libraries (embryo-endosperm DMRs) and for Col-0, Ler and Cvi embryo libraries against one another (strain DMRs). The distribution of differences highlighted global differences in methylation (*Figure 4—figure supplement 1*). To identify DMRs the analysis was restricted to differences in weighted methylation fraction of at least 35% for CG or CHG methylation, with a minimum overlap of three informative Cs between windows (i.e., at least 3 Cs at the exact same positions had sufficient coverage in both embryo and endosperm), and 10% for CHH methylation, with a minimum overlap of 10 informative Cs. We retained DMRs that had a FDR corrected p-value<0.01, reflecting whether the fraction of methylated/unmethylated counts was the same for both samples. From 365,000 to 500,000 informative 300 bp windows, 12,000–14,000 Col-Cvi and Col-Ler positive CG strain DMRs were identified, corresponding to approximately 8000 features. Most strain DMRs were in genes (*Figure 4A*). In Col-Ler comparisons we also identified 7453 features where Ler was more methylated than Col, but only 1749 features where Cvi was more methylated than Col because of the overall reduction of CG gene body methylation in Cvi (*Figure 3*, *Figure 4*, *Figure 4—figure supplement 1*). In contrast to CG DMRs, strain CHH DMRs mostly mapped to TEs and intergenic regions (*Figure 4B*), which are the sequences most likely to contain non-CG methylation. Similar results were obtained for Ler-Cvi comparisons (*Figure 4—figure supplement 2*). Independent validation by methylation-sensitive PCR or locus-specific BS-PCR of six loci confirmed our genome-wide analyses (*Figure 4—figure supplement 3*).

We identified fewer embryo-endosperm DMRs than strain DMRs, although lower BS-seq coverage in the endosperm was likely a major contributing factor (*Figure 4B*, *Figure 3—source data 1*). From 23,000 informative windows in the Col x Cvi embryo-endosperm comparison we identified 2305 positive CG DMRs, corresponding to 1100 features where the embryo was more methylated than the endosperm (*Figure 4B*). In contrast to strain DMRs and consistent with previous findings (*Gehring et al., 2009*; *Ibarra et al., 2012*), most regions hypomethylated in the endosperm in both the CG and CHH contexts correspond to intergenic regions and annotated transposable element fragments. Helitron and Mu elements were most commonly represented among demethylated TEs overlapped by a DMR, reflecting their abundance in the genome (*Figure 4—figure supplement 4*; *Gehring et al., 2009*). It is important to note that although Cvi has a global methylation profile distinct from Col and Ler (*Figure 3*, *Figure 3—source data 1*), endosperm demethylation dynamics appear to be the same. This reflects the fact that regions targeted for active DNA demethylation are depleted of genes, which is where most strain-specific methylation differences in Cvi reside.

Using our analysis pipeline we also identified embryo-endosperm DMRs from published BS-seq data from Col x Ler and Ler x Col embryo and endosperm isolated at a slightly later developmental stage (*Ibarra et al., 2012*). These datasets have higher endosperm coverage and slightly lower embryo coverage than our datasets. 40% of DMRs were in common when our union set of DMRs (identified in any matching embryo-endosperm comparison among our datasets; n = 21,973) was compared to the Ibarra embryo-endosperm DMRs (*Figure 4—figure supplement 5*). 61% of the embryo-endosperm DMRs we previously identified in Col-*gl* and Ler seeds by meDIP-seq (*Gehring et al., 2009*) were identified in our current study (*Figure 4—figure supplement 5*), indicating a high degree of overlap between this study and others.

## Embryo-endosperm DMRs are enriched for whole seed small RNAs

Because methylation of TEs and other repetitive sequences is often associated with small RNAs and because small RNAs are abundant in seeds (*Mosher et al., 2009*), we sequenced small RNAs from whole seeds at 6 DAP, obtaining 20–30 million high-quality reads for biological replicates of each

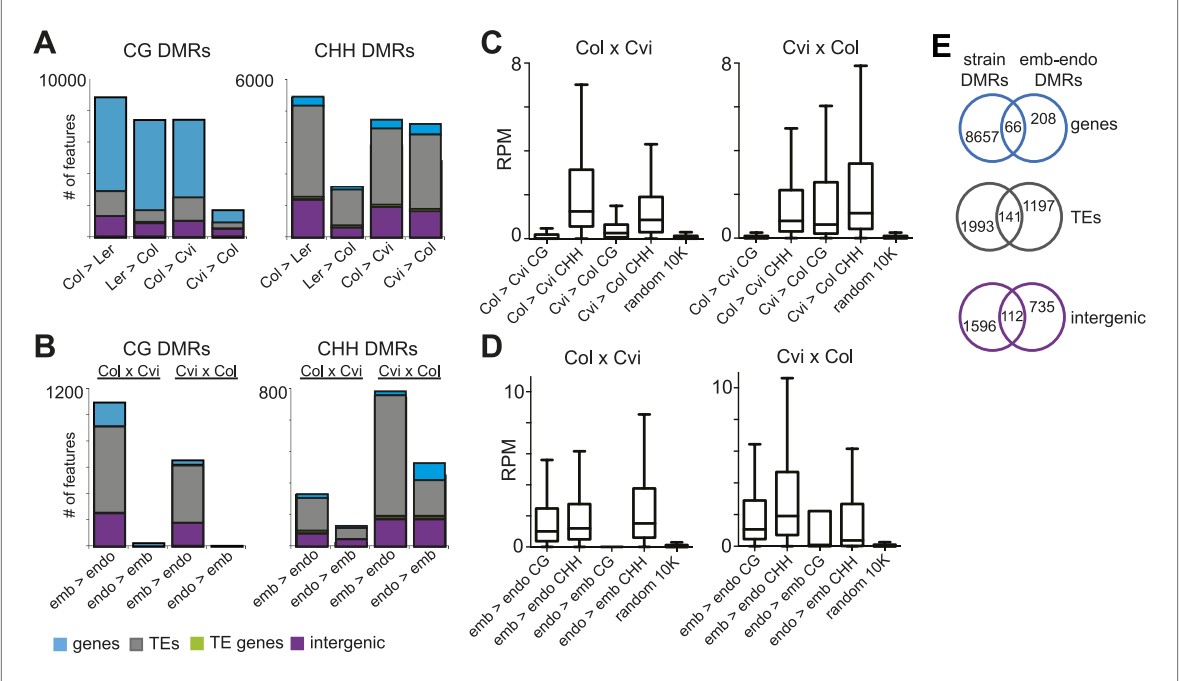

**Figure 4**. Strain DMRs and embryo-endosperm DMRs are in distinct genomic regions. (**A**) Number of features overlapping strain DMRs between Col and Ler or Col and Cvi embryos. (**B**) Number of features overlapping embryo-endosperm DMRs in Col x Cvi and Cvi x Col crosses. (**C** and **D**) 24 nt small RNA quantities (reads per million) corresponding to Col-Cvi strain (**C**) and Col x Cvi or Cvi x Col embryo-endosperm DMRs (**D**). (**E**) Overlap between Col-Cvi strain positive CG DMRs (more methylated in Col than Cvi) and the union of Col x Cvi and Cvi x Col embryo-endosperm CG DMRs (embryo more methylated than endosperm) corresponding to genes, TEs, and intergenic regions. *Figure 4—figure supplement 1* show the distribution of all methylation differences; *Figure 4—figure supplement 2* shows DMR analysis in Ler-Cvi crosses and other datasets; *Figure 4—figure supplement 3* validates DMRs identified in this analysis; *Figure 4—figure supplement 4* examines small RNAs at TEs and *Figure 4—figure supplement 5* shows the overlap of embryo-endosperm CpG DMRs with previous studies. sRNA-seq library metrics are in *Figure 4—source data 1*.

The following source data and figure supplements are available for figure 4:

**Source data 1**. Whole seed sRNA-seq libraries generated in this study.

**Figure supplement 1**. Distribution of endosperm-embryo and strain CG DNA methylation differences.

**Figure supplement 2**. Ler and Cvi strain DMRs and embryo-endosperm DMRs in additional datasets.

**Figure supplement 3**. Validation of BS-seq results with locus-specific BS-PCR or McrBC-PCR.

**Figure supplement 4**. Distribution of TE superfamilies and small RNAs within embryo-endosperm DMRs.

**Figure supplement 5**. Overlap of embryo-endosperm CG DMRs with previous studies.

sample (*Figure 4—source data 1*). Data from whole seeds cannot distinguish among small RNAs from the seed coat, embryo, or endosperm, but at 6 DAP RdDM pathway genes are expressed in both embryo and endosperm (*Jullien et al., 2012*; *Belmonte et al., 2013*), suggesting that both tissues have the ability to produce small RNAs. Strain specific CG DMRs have very low levels of small RNAs, consistent with these DMRs residing mainly in genes, which are not targeted by the RdDM pathway (*Figure 4C*). In contrast, CG DMRs where the embryo is more methylated than the endosperm are enriched for small RNAs compared to strain DMRs (Wilcoxon-Mann-Whitney test, p<0.0001) or compared to a set of random genomic loci (*Figure 4D*). TEs overlapping embryo-endosperm CG DMRs have on average higher levels of small RNAs in whole seeds than do all TEs on average from that family (*Figure 4—figure supplement 4*). Both strain and embryo-endosperm CHH DMRs are associated with small RNAs (*Figure 4C,D*).

## Correspondence between imprinted genes, differential methylation, and small RNAs

We examined the overlap of the union set of imprinted genes (including within the gene and 2 kb from the 5' and 3' ends) and the CG and/or CHH embryo-endosperm DMRs. Over 40% of the MEGs (121/285) overlapped a CG DMR within these regions, but this was not a significant enrichment compared to a random set of the same number of genes (Fisher's exact test p-value=0.3477), nor was the overlap (176/285) when Ibarra et al. CG DMRs were also included in the analysis. In contrast, PEGs were significantly enriched for CG DMRs (64/103; p=0.0174) (*Figure 5*). Furthermore, of the 29 PEGs that are in common among at least two of three sets of reciprocal crosses (*Figure 1B*), 22 are associated with a proximal TE, primarily within 1 kb 5' of the transcription start site (*Figure 5*). Fewer MEGs exhibit a correlation with presence of a TE (40/85), consistent with the lower correlation between DNA demethylation and imprinted expression for MEGs (*Figure 5*).

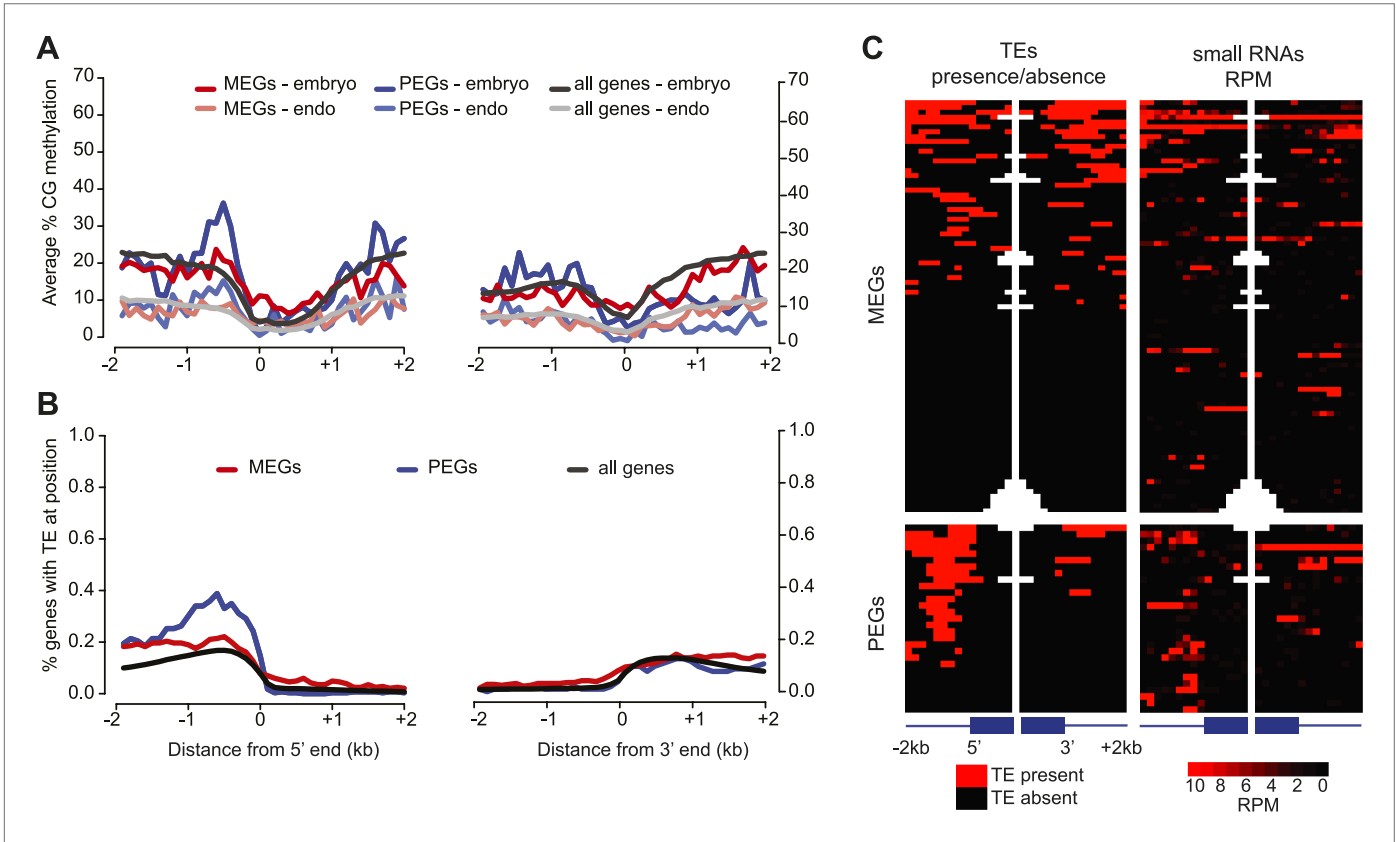

**Figure 5**. Correspondence between DNA methylation, TEs, and sRNAs for imprinted genes. (**A**) Average CG methylation in embryo and endosperm for the union set of PEGs, MEGs and all genes. (**B**) Percentage of genes with TE at indicated position. (**C**) Distribution of TEs and 24 nt small RNAs around endosperm imprinted MEGs (n = 85) and PEGs (n = 29) identified in at least two of three sets of reciprocal crosses. TE heatmap indicates the presence or absence of TEs according to TAIR10 annotation. 24 nt small RNA data is from ColxCvi whole seeds. Other libraries showed the same overall small RNA profile. Values were calculated in 200 nt windows extending 2 kb upstream and downstream from the 5' and 3' ends of the gene and 1 kb into the gene body. White indicates the absence of data. *Figure 5—figure supplement 1* shows H3K27me3 profiles around imprinted genes in vegetative tissues. *Figure 5—figure supplement 2* and *Figure 5—figure supplement 3* further explore the distribution and allelic contribution of small RNAs associated with imprinted genes.

The following figure supplements are available for figure 5:

**Figure supplement 1**. Histone H3 lysine 27 trimethylation (H3K27me3) profiles of PEGs and MEGs in vegetative tissues.

**Figure supplement 2**. Small RNA levels around imprinted genes.

**Figure supplement 3**. Fraction of maternal small RNAs near the 5' end of imprinted genes.

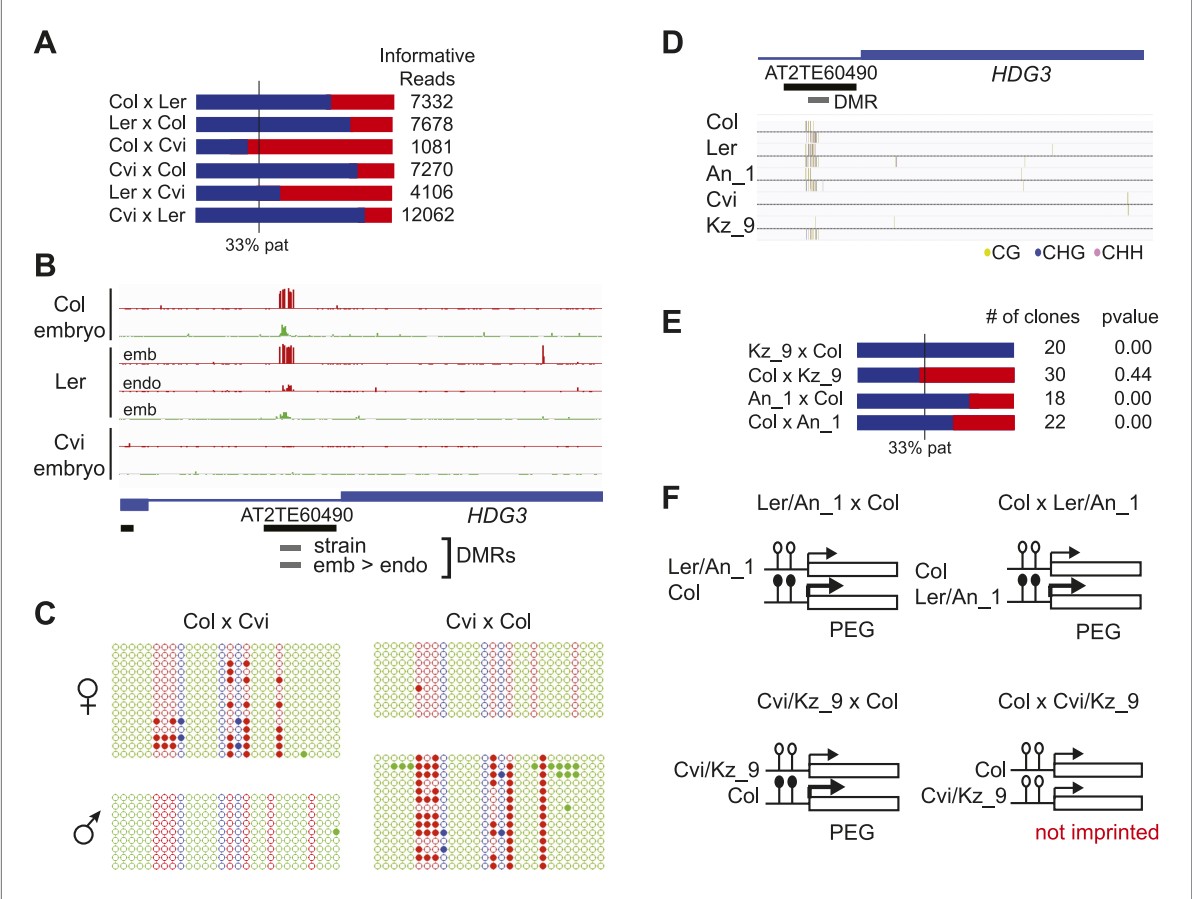

**Figure 6**. Expression and methylation analysis of *HDG3*, an allele-specific imprinted gene. (**A**) *HDG3* is a PEG except when Cvi is the paternal parent. Blue bars, % paternal allele expression; red bars, % maternal allele expression from combined mRNA-seq data; vertical line, expected percent paternal allele expression for a non-imprinted gene. (**B**) Methylation of *HDG3* 5′ flanking region in Col embryo, Ler embryo and endosperm, and Cvi embryo (additional analysis in *Figure 6—figure supplement 1*). Red track, CG; green track, CHH. (**C**) Methylation profile of maternal and paternal HDG3 alleles in Col x Cvi and Cvi x Col endosperm as determined by locus-specific bisulfite PCR. Red circles, CG; blue circles, CHG; green circle, CHH. Filled circles indicate methylation, whereas unmethylated positions are unfilled. (**D**) Methylation profile of HDG3 in Col, Ler, Cvi, Kz_9 and An_1 in leaves (http://neomorph.salk.edu/1001_epigenomes.html). (**E**) *HDG3* is not imprinted in 6 DAP endosperm when another hypomethylated strain (Kz_9) is the pollen parent, but is a PEG in a cross with another methylated strain (An_1), as determined by sequencing RT-PCR products that span informative SNPs. Blue bars, % paternal allele expression; red bars, % maternal allele expression; vertical line, expected paternal allele expression for a non-imprinted gene. The number of RT-PCR clones sequenced is indicated. p value represents a binomial test of whether the observed maternal:total ratio is less than the expected 2:3 ratio. (**F**) Cartoon representation of results. Expression and methylation results for AT3G14205 and AT2G34890 are in *Figure 6—figure supplement 2*. Examples of genetic differences causing methylation differences are in *Figure 6—figure supplement 3*.

The following figure supplements are available for figure 6:

**Figure supplement 1**. Methylation analysis of *HDG3*.

**Figure supplement 2**. Expression and methylation analysis of other variably imprinted genes.

**Figure supplement 3**. Genetic difference between strains can underlie differential methylation.

We also examined the distribution of total and allele-specific small RNAs around MEGs and PEGs. The highest average levels of small RNAs are found at the 5′ end of the gene (*Figure 5—figure supplement 2*). Despite the lack of DMR enrichment, MEGs are enriched for small RNAs within the first kilobase of the gene in all crosses examined and in the first kb 5′ of the gene in a subset of the samples (*Figure 5—figure supplement 2*). Small RNAs associated with PEGs are enriched in the first kb 5′ of

the gene—where most PEG-associated TEs are located (**Figure 5**)—and in the gene body, although levels 5′ of the gene are much higher (**Figure 5—figure supplement 2**).

We examined whether MEGs or PEGs differed in the maternal/paternal fraction of associated small RNAs. Previous data suggested that small RNAs corresponding to TEs associated with MEGs accumulate in sperm cells (**Calarco et al., 2012**); in these instances the silent paternal allele would already be targeted for RdDM in sperm. Paternal small RNAs, which in whole seeds must be derived from the endosperm or embryo genomes, constituted 6–24% of RNAs that could be assigned to a specific allele, depending on the cross (**Figure 4—source data 1**). We calculated the fraction of maternal small RNA reads for the region 1 kb 5′ and 3′ of the transcription start site of imprinted genes, retaining only those regions with at least five allele-specific reads in our analysis. MEGs were enriched for small RNAs derived from the paternally-inherited genome compared to all genes that could be evaluated for imprinting (**Figure 5—figure supplement 3**). This suggests that silencing of the paternal allele of MEGs is associated with *cis* acting small RNAs produced from those alleles in the endosperm. Small RNA data from specific compartments of the seed will be necessary to conclusively address this question.

## Allele-specific imprinted genes are associated with variable DMRs

We identified regions of the genome that are subject to DNA demethylation in endosperm in at least one background but that are variably methylated among strains. Overlap between the strain DMRs and embryo-endosperm DMRs ranged from 12% for TEs to 31% for genes (**Figure 4E**). This suggests that there are sufficient epigenetic polymorphisms at embryo-endosperm DMRs to facilitate the formation of allele-specific imprinting. Differences in imprinting among alleles tied to differences in DNA methylation could be due to genetic or epigenetic differences (**Figure 6—figure supplements 1,2, and 3**). Of the 12 allele-specific imprinted genes we identified (**Figure 2**, **Figure 2—source data 1**), 10 were associated with coincident CG or CHH embryo-endosperm DMRs and strain DMRs, 6 of which occurred at TEs (**Figure 2—source data 1**, **Figure 6**, **Figure 6—figure supplement 1**, **Figure 6—figure supplement 2**). For all 6 genes we confirmed by sequencing that the TE annotated in Col was present in the same genomic location in Ler and Cvi with no major sequences changes except for a few SNPs. These 6 genes were also included among imprinting validation assays (**Figure 1—source data 5**, **Figure 2—source data 1**). We more closely investigated three allele-specific imprinted genes with strong differences in the ratio of maternal to paternal transcripts in imprinted and non-imprinted crosses: AT2G32370, AT2G34890, and AT3G14205 (**Figure 2**, **Figure 6**, **Figure 6—figure supplement 2**). AT2G32370, *HDG3*, was originally identified as a PEG because of its association with a Col and Ler embryo-endosperm DMR and because it was expressed specifically in the endosperm (**Gehring et al., 2009**). Our new data showed that *HDG3* is not a PEG when Cvi is the male parent (**Figure 2**, **Figure 6**). The embryo-endosperm DMR associated with *HDG3* is located in a Helitron fragment (ATREP10D) 5′ of the gene, and the methylated paternal allele is predominantly expressed. This region is not methylated in Cvi (**Figure 6B**, **Figure 6—figure supplement 1**). In crosses between Cvi females and Col or Ler males, maternal and paternal alleles are differentially methylated and the gene is imprinted (the naturally hypomethylated Cvi maternal allele has the same methylation profile as an actively demethylated maternal allele). But in reciprocal crosses between Col or Ler females and Cvi males, both maternal and paternal alleles are hypomethylated and the gene is biallelically expressed in the expected 2:1 maternal:paternal ratio (**Figure 2**, **Figure 6**). This suggests that differential methylation of maternal and paternal alleles, rather than simply demethylation of the maternal allele, is required for imprinted expression. A similar logic applies to AT3G14205; the Cvi allele is hypomethylated at a 5′ RC/Helitron fragment (ATREP1) compared to Col and Ler and the gene is not a PEG when it is transmitted through the Cvi male (**Figure 2**, **Figure 2—source data 1**, **Figure 6—figure supplement 2**). AT2G34890 was a MEG except when Ler is the male parent. The Ler AT2G34890 allele is hypomethylated at a 5′ MuDR element in comparison to Cvi and Col, suggesting that loss of methylation of the paternal allele could lead to its transcription (**Figure 6—figure supplement 2**). However, consistent with a more ambiguous relationship between DNA demethylation and MEGs (**Figure 5**), closer inspection of AT2G34890 shows that at a region 5′ of the TE *both* the Ler and Cvi alleles are hypomethylated compared to Col, suggesting that a clear methylation distinction between imprinted (Col and Cvi) and non-imprinted (Ler) paternal alleles does not exist (**Figure 6—figure supplement 2**).

Allele-specific imprinted genes could represent epialleles whose expression phenotypes are observed in the endosperm. To assess how widespread allele-specific imprinting might be within the species, we determined the strain variability in methylation at regions we identified as being targeted

for endosperm demethylation, using the methylation profiling data of leaves or floral buds from 140 strains (*Schmitz et al., 2013*). This allowed us to more broadly estimate the potential for allele-specific imprinting outside of Col-Ler-Cvi strains. We divided regions corresponding to embryo-endosperm DMRs into five classes based on the range of methylation variability across all 140 strains: those with very low variability in methylation (less than 0.2 mean methylation difference across all strains), low variability (between 0.2 and 0.4 mean methylation difference across all strains), a strongly bimodal class (DMRs with a range greater than 0.4 but where most of the density of the distribution in the outer 50% of the range) and two intermediate classes, weakly bimodal and not bimodal (*Figure 7A,B*). DMRs with a high or intermediate methylation range and less strong clustering in the outer 50% of the

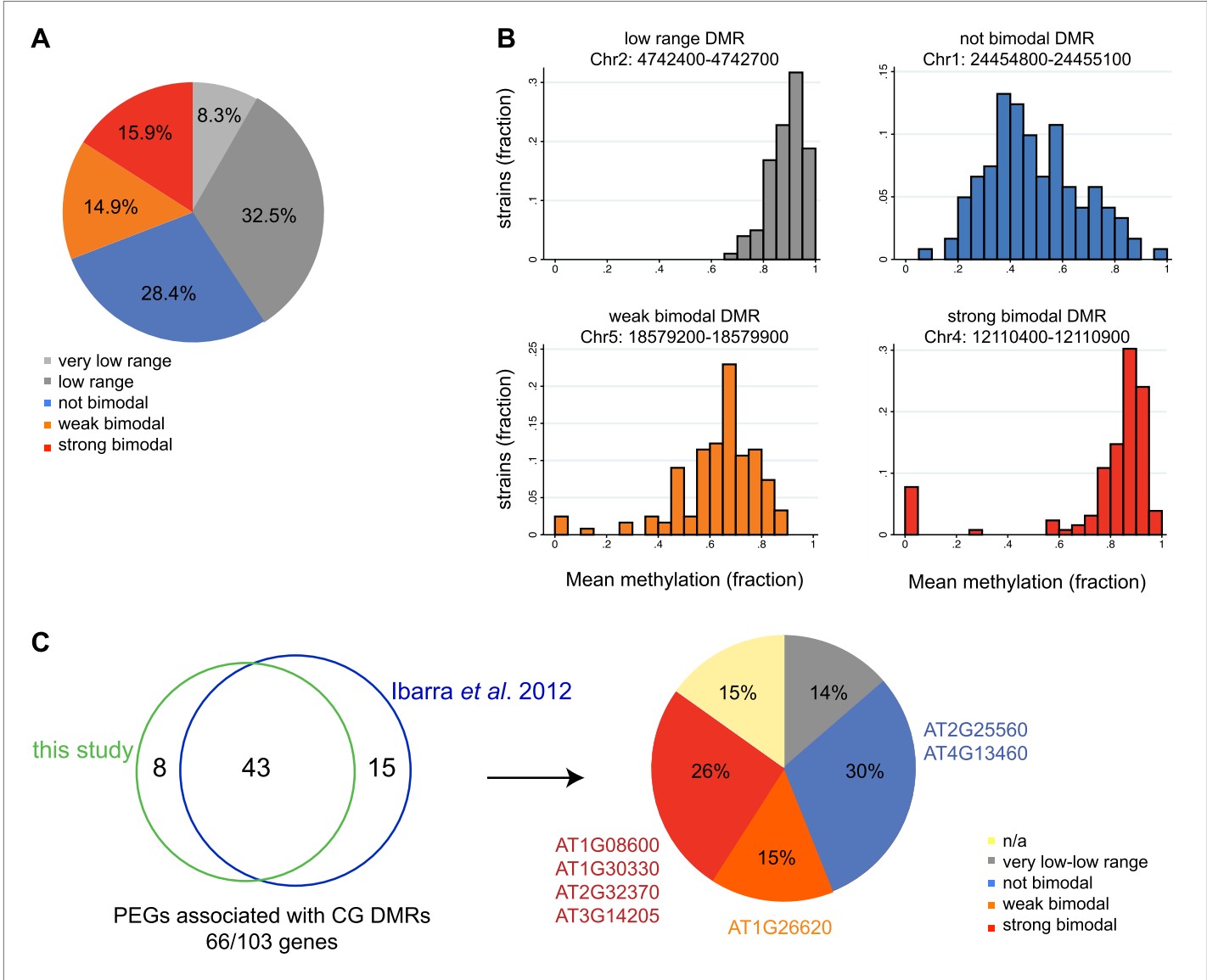

**Figure 7**. Natural epigenetic variability across strains at embryo-endosperm CG DMRs. (**A**) Methylation variability across strains for regions targeted for endosperm demethylation. Classification of methylation range in the *Schmitz et al., 2013* dataset (total of 140 strains) for all all embryo-endosperm CG DMRs (n = 10,370) identified in this study. Only DMRs with at least 5 CG sites and a minimum of five reads coverage at each site in the Schmitz et al. dataset were classified. Additionally, only DMRs with at least 70 strains with sufficient data were included. (**B**) Examples of low range (gray), not bimodal (blue), weak bimodal (orange) and strongly bimodal (red) DMRs. (**C**) Association of the classified CG endosperm-embryo DMRs with PEGs. This study: DMRs identified from all pairwise matched endosperm-embryo comparisons from bisulfite datasets in *Figure 3—source data 1*; *Ibarra et al., 2012*: ColxLer and LerxCol DMRs combined. Allele-specific PEGs are listed around the pie chart. n/a = not classifiable because gene was associated with DMR of more than one type.

distribution were considered weakly bimodal. All remaining DMRs were classified as 'not bimodal'—these regions tend to have a more uniform or unimodal distribution across a large range of scores (*Figure 7A,B*). The majority of DMRs (69%) were consistently methylated across all strains examined, falling into the very low range, low range, or not bimodal categories (*Figure 7A,B*). However, about 16% of DMRs were in the strongly bimodal category, where methylation is consistently high or low across strains except for one or more strong outliers (*Figure 7A,B*).

We overlapped our union set of imprinted genes with the classified DMRs. 21 MEGs and 9 PEGs were associated with a very low or low variability DMR within the gene or two kb 5′ or 3′. These include the MEG *FWA*, a locus with high levels of promoter methylation in all strains. We thus expect that in all strains in the endosperm *FWA* will have a demethylated maternal allele and a highly methylated paternal allele, and be consistently imprinted within the species. In contrast, 27 MEGs and 17 PEGs (11% of all imprinted genes) were associated with strongly bimodal DMRs, including four PEGs we identified from our Col, Ler, and Cvi mRNA-seq and BS-seq data as exhibiting allele-specific imprinting and being associated with a shared embryo-endosperm and strain DMR (*Figure 7C*, *Figure 2—source data 1*). This includes the DMRs associated with the allele-specific imprinted genes *HDG3* and AT3G14205 (*Figure 7C*).

We tested whether the methylation status of the embryo-endosperm DMRs associated with allele-specific imprinted genes was predictive for imprinting. If the difference in imprinting among *HDG3* alleles was due to the *cis* epigenetic difference at the 5′ TE, we predicted that crosses with other male parents carrying naturally hypomethylated alleles would exhibit lack of imprinting in $F_1$ endosperm. The strain Kz_9 has reduced methylation at the 5′ TE, although it is more methylated than in Cvi (*Figure 6D*). We performed reciprocal crosses between Col and Kz_9, extracted the RNA from endosperm 6 DAP and performed RT-PCR around a Col/Kz_9 SNP. Cloning and sequencing of the PCR products confirmed that *HDG3* was a PEG when Col was the male parent. In the reciprocal cross, when the hypomethylated *HDG3* allele was inherited from the Kz_9 male, *HDG3* was not imprinted (*Figure 6E,F*). *HDG3* remains imprinted in crosses between Col and another strain with methylation, An_1 (*Figure 6D–F*). This suggests that the epigenetic state of the *HDG3* allele is sufficient to predict imprinting in other naturally occurring strains and is thus likely causal for lack of imprinting in Cvi. Similarly, the DMR variability analysis (*Figure 7*) identified another strain, Seattle_0, where the TE 5′ of AT3G14205 was hypomethylated. Similar to crosses with Cvi males, in crosses where Seattle_0 was the male parent, AT3G14205 was no longer a PEG (*Figure 6—figure supplement 2*), but was still a PEG when Seattle_0 was the female and Col the male. As expected for AT2G34890, where the methylation state of the 5′ region was variable but did not seem to correlate with whether the allele was imprinted or not among Col, Ler, and Cvi, we found that AT2G34890 is a MEG in crosses with Col regardless of whether Es_0, a strain with a hypomethylated allele like Ler, served as the female or the male parent (*Figure 6—figure supplement 2*).

As we have shown for *HDG3* and AT3G14205, crosses involving a methylated allele and a naturally hypomethylated allele can result in allele-specific imprinting in one direction of the cross. Thus, in addition to the allele-specific imprinted genes we identified based on our mRNA-seq data, the additional MEGs and PEGs associated with embryo-endosperm DMRs that are strongly bimodal for methylation within the population might not be imprinted in strains where the allele is naturally hypomethylated. This suggests that 11% of imprinted genes have the potential to be variably imprinted when a particular strain is the male or female parent.

## Discussion

By examining crosses among three of the thousands of Arabidopsis strains, we were able to identify genes imprinted among all crosses and a small number of strong candidates for allele-specific imprinting caused by a *cis* methylation difference at a proximal transposable element. We found that most imprinted genes identified in reciprocal crosses between two strains are imprinted in crosses between other strains, or have some evidence for parental bias even if all imprinting criteria were not met. Arabidopsis allele-specific imprinted genes represent a small fraction of all possible imprinted genes, approximately 6% in our mRNA-seq datasets. These results are consistent with the extent of allele-specific imprinting in maize, where around 12% of imprinted genes fall into this category (*Waters et al., 2013*). Allele-specific imprinting could represent genes that have recently come under the control of a TE and may or may not have an endosperm function, or genes that are imprinted specifically in one strain and not the other due to differences in control over endosperm growth and development.

Under the conflict theory (**Haig, 2013**), differences in seed size could reflect strains that have reached different optima for imprinting—strains producing small seeds could be considered more maternalized and those producing larger seeds more paternalized. Arabidopsis allele-specific imprinted genes encode multiple types of proteins, including a putative transcription factor (*HDG3)*, a heat shock protein, a phosphoinositide phosphatase (AT3G14205), three chromatin proteins, and a gene required for normal levels of phytic acid accumulation in seeds (**Kim and Tai, 2010**). Further experimentation exploring the function of these genes during seed development in each genetic background will be required to determine if differences in imprinting among strains contribute to seed phenotypes.

By considering population methylation variation at regions demethylated during reproductive development, we were able to estimate the possible extent of allele-specific imprinting within the species. Of the set of 388 imprinted genes we identified, 11% were associated with embryo-endosperm DMRs that are variably methylated in vegetative tissues in at least one of 140 strains (the strongly bimodal DMRs). Based on their methylation patterns, these genes have the potential to act as epialleles and exhibit allele-specific imprinting, as we demonstrated for *HDG3* and AT3G14205. However, we caution that imprinted genes associated with a strongly bimodal DMR do not always exhibit allele-specific imprinting when alleles with different methylation patterns are combined in $F_1$ endosperm. For example, we originally identified AT4G23110 as a candidate imprinted gene because it was associated with an embryo-endosperm DMR and only expressed in seeds (**Gehring et al., 2009**), which was later confirmed by mRNA-seq (**Gehring et al., 2011**). In this study we find that the embryo-endosperm DMR in the 5′ region of the gene is strongly bimodal within the population, with 118 strains exhibiting between 60–100% CG methylation in a 500 bp 5′ region, and 10 strains with no methylation. However, when we performed reciprocal crosses between one of the hypomethylated strains and Col, a methylated strain, imprinting was maintained in both directions of the cross. Thus, we expect that 11% is the maximal fraction of genes that would exhibit allele-specific imprinting due to strain-to-strain differences in methylation, with the actual number being less than this.

The predictive power for whether a gene will exhibit allele-specific imprinting is greater for PEGs than for MEGs. PEGs were more often associated with embryo-endosperm DMRs than MEGs, and showed stronger correlation with the presence of a TE, and the presence of H3K27me3 in vegetative tissues (**Figure 5—figure supplement 1**; **Lafos et al., 2011**). Our data and that of many others (**Köhler et al., 2012**; **Gehring, 2013**; **Zhang et al., 2014**) suggest that the mechanism of imprinting for PEGs in Arabidopsis and maize is that the methylated paternal allele is expressed while the hypomethylated maternal allele is silenced by the PRC2 complex. Thus, the imprinting state of PEGs is much easier to predict based on known genetic and epigenetic characteristics. MEGs represent a more diverse class of genes, and their imprinting status is more difficult to predict based on DNA methylation or TE presence, although we do find that MEGs are enriched for small RNAs in seeds.

The RC/Helitron class of TEs, the most abundant type in euchromatin, is prevalent among DMRs. RC/Helitrons are also one of the most variably methylated TEs, with more than half being poorly or unmethylated in Col (**Ahmed et al., 2011**). The susceptibility of Helitrons to loss of methylation coupled with their overabundance among embryo-endosperm DMRs further suggests that imprinting could be highly dynamic if considered on a population scale. McClintock described the ability of TEs to cycle in nature and between active and inactive epigenetic states (**McClintock, 1965**; **Slotkin and Martienssen, 2007**). A conceptually similar phenomenon could cause imprinting to vary between closely related genotypes.

Our study also yielded several unanticipated findings. We discovered that the Cvi strain is globally hypomethylated, and primarily loses CG methylation in gene bodies. Despite its ubiquity, the function of CG gene body methylation in plants and animals is still unclear, although it may be important for regulating alternative splicing or preventing spurious transcription (**Zilberman et al., 2007**; **Shukla et al., 2011**). Mutations that reduce CG gene body methylation (e.g., mutations in the maintenance DNA methyltransferase) affect CG methylation in all genomic contexts. Because Cvi has little gene body methylation but nearly normal levels of CG methylation in other genomic features, Cvi could serve as a model genotype to further explore the function of gene body methylation in relation to gene expression or chromatin structure. Cytologically, Cvi has reduced heterochromatin and dispersed 45S rDNA repeats and DNA methylation compared to Col and Ler (**Tessadori et al., 2007**, **2009**). Decondensed chromatin might facilitate increased production of small RNAs (**Schoft et al., 2009**), and Cvi x Col endosperm showed little reduction in CHH methylation compared to the embryo. Consistent with this, small RNAs produced from the maternal allele represented a greater fraction of small RNAs

when Cvi was the female parent in a cross (*Figure 4—source data 1*). We also observed, in contrast to a previous study (*Mosher et al., 2009*), that small RNAs derived from the paternally inherited genome were readily detected in seed tissue from all crosses. As expected, maternal small RNAs were much more abundant than paternal small RNAs, but because we could not distinguish the three components of the seed (embryo, endosperm, and seed coat), it is unknown whether this was due to predominantly maternal expression in the endosperm, or reflective of the overall maternal bias expected in whole seeds. However, the fraction of paternal small RNAs (6–24%) was comparable to the fraction of paternal reads when mRNAs were sequenced from whole seeds (15–20%) (*Figure 1—source data 1*, *Figure 4—source data 1*), perhaps suggesting that small RNAs in seeds reflect the ratio of maternal:paternal genomes in each tissue. Our analysis of the previously published data from Col x Ler and Ler x Col siliques (*Mosher et al., 2009*) also revealed the presence of paternally derived small RNAs, although at lower levels than in our whole seed datasets (*Figure 4—source data 1*). Recent analysis of small RNAs in rice inter-strain crosses detected maternally and paternally derived small RNAs in the endosperm and suggested rice might be different from *A. thaliana* in this regard (*Rodrigues et al., 2013*). Our data instead suggest that rice and Arabidopsis are likely similar in terms of parental small RNA composition in the endosperm, although endosperm-specific Arabidopsis profiles will be necessary to conclusively evaluate this. Interestingly, at 6 DAP the largest subunit of *Pol IV*, *NRPD1a*, was paternally biased in all crosses. *NRPD1a* has a 5′ embryo-endosperm DMR associated with small RNAs, further suggesting that there may be a complex interplay and crosstalk between active DNA demethylation and the small RNA production pathway. Other genes that promote DNA methylation were also primarily paternally expressed in all crosses (e.g., the *VIM* family genes). A major unsolved question is how the maternal alleles of genes that are demethylated before fertilization remain hypomethylated several days after fertilization, in the spite of the presence of small RNAs that normally target DNA methylation to these sequences. It is possible that changes to maternal endosperm chromatin structure could alter the efficacy of RNA-directed DNA methylation.

In conclusion, our study demonstrates that epiallelic variation and genomic imprinting mechanisms intersect to produce novel gene expression patterns in seeds. We propose that the phenotypic impact of epialleles is likely to be most pronounced in the endosperm because changes in DNA methylation are a programmed part of endosperm development. Epialleles naturally circulating in plant populations might significantly impact seed development and lead to the production of novel phenotypes in hybrids.

## Materials and methods

### Plant material

Plants were grown in a greenhouse with 16-hr days at ~21 C. Flowers were emasculated 2 days before pollination. Seeds were dissected at 6 DAP, which corresponds to the torpedo stage of embryogenesis under our growth conditions (*Figure 1—figure supplement 1*).

### mRNA-seq library construction

RNA was isolated from endosperm, embryo, and whole seeds 6 days after pollination as described (*Gehring et al., 2011*) using either the RNAqueous kit with Plant RNA Isolation Aid or the RNAqueous Micro Kit (Ambion, Life Technologies Corporation, Carlsbad, CA). At least 600 ng of DNAse I-treated RNA (Invitrogen, Life Technologies Corporation, Carlsbad, CA) was used to prepare mRNA-Seq libraries as described (*Gehring et al., 2011*), except that Illumina TruSeq primers were used in the final amplification step. Amplification was for 12 cycles. Strand-specific RNA-seq libraries were generated from at least 100 ng of total RNA using the Integenex PolyA prep protocol (Wafergen Biosystems, Fremont, CA) with 15 cycles of amplification. See *Figure 1—source data 1* for details of library prep for specific samples.

### mRNA-seq data analysis

Endosperm mRNA-seq was performed in triplicate (18 samples). Because we previously found little evidence for imprinting in embryos at this stage of development (*Gehring et al., 2011*), embryo mRNA-seq was performed on single samples (*Figure 1—source data 1*). Single-end sequencing of mRNA-Seq libraries was performed on an Illumina HiSeq machine. Read length was 40 bp or 80 bp (*Figure 1—source data 1*). Sequencing quality was assessed using fastqc, and raw reads were filtered for overrepresented adapter sequences using fastx_clipper. Low quality reads were removed with fastq_quality_filter using the options –q 20 and –p 80 (http://hannonlab.cshl.edu/fastx_toolkit/). Filtered

reads were then aligned to the TAIR10 version of the Arabidopsis genome using Tophat v2.0.8 (*Trapnell et al., 2009*). For 40 bp libraries the options -solexa1.3-quals --segment-length 18 --segment-mismatches 1 --max-segment-intron 11,000 were used and for 80 bp libraries the options were --solexa1.3-quals --segment-length 30 --max-segment-intron 11,000. Reads counts for each gene and TE annotated in TAIR10 were quantified using htseq-count (http://www-huber.embl.de/users/anders/HTSeq/doc/index.html) with options –m intersection-strict –-stranded = no (for non-strand-specific libraries) or –stranded = yes (for strand-specific libraries). After sorting reads by genome position, single nucleotide polymorphisms (SNPs) (*Supplementary file 2*) were used to classify reads by strain using a custom script (*Supplementary file 3*). Reads were discarded if classification at two SNP positions within the same read conflicted. Htseq-count with the same options was run on each allele-specific set of mapped reads to generate allele counts for each gene and TE. Libraries ranged in depth from 50 to 160 million high-quality reads per sample, of which approximately 4–10 million reads could be assigned to a specific parental allele (informative reads). The highest proportion of informative reads was in crosses between Ler and Cvi, which have the most SNPs (*Figure 1—source data 1*).

### Identifying imprinted genes

We slightly modified our previously published analysis method (*Gehring et al., 2011*) to identify imprinted genes. We used Fisher's exact test on each set of reciprocal crosses to test the null hypothesis that $p_1 = 2p_2 = 0.67$ ($p_1$ = portion of strain A reads in A female × B male and $p_2$ = portion of strain A reads in B female × A male) for endosperm or $p_1 = 2p_2 = 0.5$ for embryo. We considered genes with a Benjamini corrected p value less than 0.01. We further filtered the list by removing genes with an imprinting factor (*Gehring et al., 2011*) less than 2 and by removing genes that were more than twofold higher expressed in the seed coat than embryo or endosperm at the linear cotyledon stage, using data from *Belmonte et al. (2013)*. To obtain values from the Belmonte et al. data, for endosperm we averaged the RMA values from the MCE, PEN, and CZE samples and for seed coat we averaged values from the CZSC and SC, all from the linear cotyledon stage. Averaged values were log transformed and genes with a seed coat-endosperm or seed coat-embryo differences $\leq$1 retained. Finally, for maternally biased genes we required that at least 85% of informative reads were maternal in both directions of the reciprocal cross and for paternally biased genes that at least 50% of informative reads in both directions of the reciprocal cross were paternal. For embryo libraries the final filtering step required at least 70% of reads to be maternal in both directions of the cross for maternally biased genes and less than 30% maternal for paternally biased genes. For each set of endosperm reciprocal crosses we sequenced mRNA from three biological replicates. For a gene to be called imprinted in a particular set of endosperm crosses (Col-Ler, Ler-Cvi, or Col-Cvi), it had to be called imprinted in 2 of 3 reciprocal cross pairs of biological replicates. We chose two of three instead of three of three because of variation in sequencing depth among libraries.

### Identifying allele specific imprinting

We calculated the ratio of (maternal reads$_{(strain A × strain B)}$ + 1)/(paternal reads$_{(strain A × strain B)}$ + 1) and plotted it against the ratio of (maternal reads$_{(strain B × strain A)}$ + 1)/(paternal reads $_{(strain B × strain A)}$ + 1) for all loci with >0 informative reads and seed coat-endosperm expression $\leq$1 (*Belmonte et al., 2013*). The Euclidean distance of every point to the lines x = 2 and y = 2 (no imprinting) was calculated using MATLAB. Loci within the distance of 1 to each line were retained for further analysis. For each locus we also calculated a parental bias factor, b(g) (b(g) = log$_2$ (maternal reads + 1) − log$_2$ (2 × paternal reads + 1)) and a normalized parental bias factor b$_{norm}$(g) = b(g) − mean(b(g))/stdev(b(g)). For non-imprinted genes the value of b(g) = 0 and b$_{norm}$(g) = 0. Loci within 1 of x = 2 or y = 2 that also fell above the 95th (maternal bias) or below the 5th (paternal bias) percentile distribution of b$_{norm}$(g) values exhibited parental bias in only one set of reciprocal crosses. Loci that showed maternal or paternal preference when a particular strain was the parent were determined by intersecting lists from each cross (e.g., intersect Col-Ler maternal biases with Col-Cvi maternal biases to generate a list of loci that are maternally biased when Col is the female parent). To generate the final list of genes that exhibit potential allele-specific imprinting, the loci described above were intersected with the list of imprinted loci from the reciprocal crosses not involving the strain exhibiting the bias (e.g., intersection of loci maternally biased when Col is the male parent but not when it is the female parent with MEGs in Ler-Cvi reciprocal crosses).

### Validation of imprinted genes

Validation of the global RNA-seq analysis was by amplicon sequencing of RT-PCR products using miSeq or Sanger sequencing, or by CAPs digestion. RNA was collected from dissected embryo and

endosperm, reverse transcribed, treated with DnaseI, and amplified with ExTaq for 30 cycles (PCR primers in *Supplementary file 1*). For miSeq analysis, amplicons from the same cross and tissue type were pooled and libraries were constructed using the Illumina NexteraXT kit (Illumina, Inc., San Diego, CA). Paired-end sequencing of RT-PCR amplicons was performed on an Illumina MiSeq machine, generating 150 bp reads. Raw reads were aligned to a pseudometagenome (see small RNA methods) using Tophat v2.0.8 alignment options –read-mismatches 10 –edit-distance 12. After sorting reads by genome position, SNPs (*Supplementary file 2*) were used to classify reads by strain just like for mRNA-seq. A goodness-of-fit binomial exact test was used to test the null hypothesis that the observed fraction of reads derived from the maternal allele for each assayed locus was described well by the binomial distribution parameterized by p = the observed fraction maternal reads from the combined counts of the RNA-seq libraries analyzed in this study.

## GO analysis

GO analysis was performed using the DAVID bioinformatics resource version 6.7 (*Huang et al., 2009a*, *2009b*). Reported p-values are corrected using the Benjamini method.

## Whole genome bisulfite sequencing library construction

Whole genome bisulfite sequencing library construction and data analysis are described in detail at Bio-protocol (*Pignatta et al., 2015*). Seeds from 30 to 40 siliques per sample were dissected and DNA was extracted as described (*Gehring et al., 2009*). At least 1 µg of RNAse-treated DNA was used for library preparation (*Figure 2—source data 1*). Libraries were made using an Illumina TruSeq kit (Illumina, Inc.), with the following modifications. DNA was sheared using a Covaris instrument (settings: peak power 175 W, duty factor 10, cycles/burst 200, time 6 min, 6 C), purified using Agencourt AMPure beads (1.4× DNA:beads) (Beckman Coulter, Inc., Brea, CA), and resuspended in 50 µl of water followed by end repair and 3′ end adenylation. Illumina TruSeq DNA adapters, which contain 5-methylcytosines instead of cytosines, were ligated in a 50-µl overnight reaction at 16°C with 2.5 µl adapters, 5000 units T4 DNA Ligase (New England BioLabs, Ipswich, MA), and 1 × T4 Ligase buffer with ATP. DNA was cleaned twice using Agencourt AMPure beads before bisulfite treatment with the MethylCode Bisulfite Conversion Kit as per manual instructions (Invitrogen, Life Technologies Corporation). Bisulfite-treated DNA was eluted in 10 µl. Three µl were used as a template in each of two PCR reactions with 0.5 units Pfu Turbo, Cx Hotstart DNA Polymerase (New England BioLabs), 1 µl 10 mM dNTPs, and 1 × Turbo Cx buffer. PCR conditions were: 95°C for 2 min, 12–15 cycles (95°C for 20 s, 60°C for 30 s, 72°C for 1 min), and 72°C for 7 min. Libraries were subjected to QC on a bioanalyzer before sequencing on a Illumina HiSeq2000 using a single read 80 base pair protocol except for one library which was sequenced using 2 × 100 paired end reads.

## Whole genome bisulfite sequencing data analysis

Adapters and low quality reads (less than 75% quality scores above 25) were discarded after running quality control of sequencing reads with fastqc (http://hannonlab.cshl.edu/fastx_toolkit/). Libraries prepared from Col and Cvi were aligned to TAIR10 genome using Bismark (*Krueger and Andrews, 2011*) with the following parameters: -n 1 -l 50, where n is the maximum number of mismatches and -l the length of seed (first number of nt that are mapped with less than n mismatches). For Cvi, reads that failed to map to TAIR10 were then mapped against the Cvi pseudogenome (TAIR10 genome with Cvi SNP substitutions). All mapped reads were combined. Libraries prepared from Ler were aligned to the Ler-0 genome (*Gan et al., 2011*). After mapping, a 2-strain alignment was used to convert Ler-0 genome coordinates to TAIR10 coordinates, allowing subsequent pairwise comparisons between libraries. Redundant mapped reads were eliminated from each library starting from a sorted SAM file, keeping only one sequence per strand that mapped to the same position. To do this, reads were sorted by decreasing prevalence followed by increasing number of mismatches (not counting bisulfite conversions) to the genome. The most prevalent read with the total highest quality string was kept. In the case of a tie, the read with the fewest number of mismatches was retained. Bismark's methylation extractor script was used to calculate a methylation value for each cytosine.

For $F_1$ hybrid libraries, we first mapped the reads to one of the parental genomes (Col-Cvi reads to TAIR10 and Ler-Cvi reads to Ler). Reads that failed to map to either the TAIR10 or Ler genomes were mapped against the Cvi pseudogenome using the same alignment parameters (-n 1 -l 50). We discarded redundant reads and combined the remaining Cvi-mapped reads to the ones already mapped against Col-0 (TAIR10) or Ler. To assign reads to a particular strain and to retain as many unique reads as possible, we separated the reads by strand and ignored C>T SNPs for forward reads and G>A SNPs

for reverse reads. Reads were classified as maternal, paternal, no evidence for either genome (for reads not overlapping any SNP), or both (conflicting data) based on their sequence at known SNP positions. After classification, redundant reads from each class were eliminated as described above, and methylation extractor was run for each class as well as for all reads combined. The mean bisulfite conversion rate for each library was calculated based on the methylation status of each cytosine from reads mapping to the chloroplast genome, which are expected to be unmethylated.

## Identifying differentially methylated regions (DMRs)

Bismark's methylation extractor output was summarized by chromosome position by converting the methylation string into ummethylated counts, methylated counts, and percent methylation. The genome was divided into 300 nt windows that overlapped by 100 nt. Using Bismark's methylation extractor output files as input, weighted methylation levels for each window were calculated as described (*Schultz et al., 2012*), with the requirement of at least 5-read coverage at each site. Differential methylation was assayed by calculating the difference (sample 1 − sample 2 of weighted methylation fractions) and confidence (p-value from Fisher's exact test) for each window in all sequence contexts. p values were corrected with the Benjamini and Hochberg False discovery rate (FDR). We defined CG and CHG DMRs as windows with a weighted methylation difference of at least 35, with a minimum overlap of three informative Cs between windows and a corrected p value<0.01. CHH DMRs had a weighted methylation difference of at least 10, with a minimum 10 overlapping informative cytosines and a p value<0.01.

To compare methylation levels between strains, methylation fractions of sites in genes and TEs were scaled relative to the mean level of the Col strain in that context. Following analysis by ANOVA, the magnitudes of differences in the normalized methylation levels between strains were calculated using Tukey's HSD test, with alpha set at 0.05 (*Figure 3—source data 2*).

To visualize methylation profiles around imprinted genes, the methylation level of each assayed position (with at least five informative reads) was summarized into a set of bed files and used as input to calculate the average methylation in 200 nt windows spanning extended gene bodies (from 2 kb upstream of the transcription start to 2 kb downstream of the transcription end) of the conserved imprinted genes. We also compared Col, Ler, and Cvi embryo CG average methylation using 50 nt windows spanning extended bodies of genes and TEs. Using a set of genome features of a specific type (e.g., list of genes, TEs), we summarized the methylation level across that set of features. We calculated the weighted mean methylation (wmean) for a particular region (R) containing n sites (i) with methylation data (me_i) with the equation: $wmean(R) = \Sigma[i = 1 \text{ to } n](me\_i/tot\_i)$ where tot_i is the total number of reads. The results of these analyses were displayed using R (*Figure 5*). Plots for *Figure 3* and *Figure 3—figure supplement 1* were also generated using R.

## Validation of BS-Seq results with locus-specific BS-PCR or McrBC-PCR

DNA from embryo, endosperm and leaves was extracted using a CTAB protocol. Bisulfite treatment was performed using the MethylCode Bisulfite Conversion Kit (Invitrogen, Life Technologies Corporation) following the manufacturer's protocols. PCRs were performed using the primer pairs listed in *Supplementary file 1*. PCR products were gel purified, cloned and sequenced. Sequences were aligned using SeqMan and methylation was measured using Kismeth (*Gruntman et al., 2008*). For validation using McrBC, approximately 800 ng of genomic DNA were digested overnight at 37°C with 50 U of McrBC (New England Biolabs) in a 30 µl reaction. 5 µl were used as template in PCR along with an untreated DNA sample. Primer pairs are listed in *Supplementary file 1*.

## Assessing DMR variability among 140 strains

To determine how variably the identified embryo-endosperm DMRs were methylated across various wild-type strains, we downloaded BS-seq data from NCBI GEO accession GSE43857 (*Schmitz et al., 2013*). For strains with both leaf and bud data, only the leaf data was used, resulting in data for 140 different Arabidopsis strains. Methylation data were extracted from the Schmitz et al. data for all of our embryo-endosperm DMRs; contiguous or overlapping DMRs were merged prior to the analysis. For each DMR we assigned a methylation score equal to the weighted mean methylation at each CpG site in the DMR (*Schultz et al., 2012*) for each of the 140 strains. Only CpG sites with at least five reads of support and only strains with five or more CpG sites for a DMR were included in the analysis. DMRs were censored from the analysis if more than 70 of the 140 strains had missing scores. The remaining DMRs (n = 10,370) were classified into five categories that describe the distribution of methylation across the Schmitz et al. strains. DMRs with a low range of methylation scores, defined as a difference of less than 0.4 between the scores

of the most and least methylated strains, were considered to have roughly consistent methylation patterns across all strains. This was further divided into a 'very low range' subcategory consisting of all DMRs with a range of methylation scores less than 0.2 across all strains. These DMRs have very consistent methylation scores across all strains, and tend to correspond to highly methylated transposable elements or regions near centromeres. DMRs with a score range greater than 0.4 were further subdivided according to whether the strain scores tended to be unimodal or uniformly distributed across the range of the data, or whether they tended to be bimodally distributed. This was determined by counting the number of strains whose methylation scores fell in the middle 50% of the data range, and comparing this to the number of strains whose methylation scores fell in the top or bottom 25% of the range. DMRs with a high range but with most of the density of the distribution in outer 50% of the range of the data were considered 'strongly bimodal' (range >0.7 and fraction strains in upper or lower 25% of the range ≥0.8). These DMRs include cases where methylation is consistently high or low across strains except for one or two strong outliers. DMRs with high or intermediate range and less strong clustering in the outer 50% of the distribution were considered 'weakly bimodal' (range >0.7 and fraction in outer 50% ≥0.5 or 0.7 ≥range ≥0.4 and fraction in outer 50% ≥0.8). All remaining DMRs were classified as 'not bimodal'—these tend to have a more uniform or unimodal distribution across a large range of scores.

## Small RNA library construction

RNAs less than 200 bp were isolated from whole seeds 6 DAP using the miRVana RNA isolation kit (Ambion, Life Technologies Corporation). Libraries for Illumina sequencing were constructed following the method of *Grimson et al. (2008)*, with only minor modifications. Instead of $^{32}$P-labelled oligos, 18 nt and 30 nt unlabeled marker RNAs were used in conjunction with SYBR-Gold for size selection and monitoring of ligation reactions. Marker RNAs were kept in separate lanes on the polyacrylamide-urea gels instead of being directly mixed with the RNA samples, but were processed in an identical manner. We enabled multiplexing of libraries by using four different 3' PCR primers (*Supplementary file 1*) during library amplification, each of which was 94 bases in length as opposed to the 44 nt 3' PCR primer from the referenced protocol. As a result, the sequences obtained in the final gel purification step ranged in size from 135 to 155 nt in length.

## Small RNA analysis

Single-end sequencing of sRNA libraries was performed on an Illumina HiSeq machine (four libraries of 40 bases per lane). We trimmed low-quality read ends (with fastq_quality_trimmer –t 20 and –l 25) and removed adapters (fastx_clipper tool –a TCGTATGCCGTCTTCTGCTTG –i 18; http://hannonlab. cshl.edu/fastx_toolkit/). Reads were aligned using Bowtie 1.0.0 (*Langmead et al., 2009*) using the parameters –v 2 and --best, such that up to two mismatches were allowed and any read mapping to multiple locations was randomly assigned to one of the locations that had the best match to the read. We used the resequenced Ler genome (*Gan et al., 2011*), and TAIR10 for the Col genome. We constructed a Cvi pseudogenome in which Cvi SNPs and 1 bp indels (obtained from http://signal.salk.edu/ atg1001/download.php) were used to modify the TAIR10 genome at the appropriate positions. To facilitate unbiased mapping, sRNA reads from hybrids were aligned to a metagenome composed of the two parental genomes. The reads were then converted to TAIR10 coordinates, regardless of the parent strain of origin, and were classified using the same SNP classification approach described for the mRNA-Seq analysis. All reads that overlapped annotated tRNAs, snRNAs, rRNAs, or snoRNAs were removed. We normalized libraries by converting read values within windows into RPM (reads per million) values. The conversion from reads to RPM used the total number of reads aligning to the genome for each library following the subtraction of structural RNAs.

## Generating SNP lists

To generate a Col/Ler SNP list, we used the previously described Col/Ler SNP list (*Gehring et al., 2011*) supplemented with novel SNPs having an unambiguous consensus base (A,C,T,G), PHRED ≥ 25, detection score = 1 from the resquenced Ler genome (*Gan et al., 2011*) (http://mus.well.ox.ac. uk/19genomes/variants.SDI/). In total, this yielded a list of 384,612 SNPs. To generate an initial Col/Cvi SNP list, we downloaded the SALK Cvi_0 (accession CS28198) data from http://signal.salk.edu/ atg1001/download.php. From the quality_variant_filtered_Cvi_0.txt file, we removed SNPs with <0.95 or less concordance (n = 66,272) as well as 1 bp deletions (n = 11,444). This yielded 579,310 remaining SNPs. We derived an initial Ler/Cvi SNP list using the following logic: (1) if the SNP was present in Col/ Cvi list but not Col/Ler list, the Col/Cvi SNP was added to the Ler/Cvi list; (2) if the opposite scenario

was true (present in Col/Ler but absent from Col/Cvi), the Col/Ler SNP was inverted (e.g., C>T becomes T>C) and added to the Ler/Cvi SNP list; (3) if the same SNP was found in both Col/Ler and Col/Cvi SNP lists, it was not added to the Ler/Cvi list since these SNPs arise from a difference between Col and both Ler and Cvi genomes. This yielded a list of 645,212 SNPs. In order to assess the rates of erroneous read classification using these SNP lists, we used them to classify reads of known origin from mRNA-seq and small RNA-seq libraries made from Col x Col, Ler x Ler, and Cvi x Cvi embryo and endosperm or whole seeds. SNPs that systematically misclassified reads were filtered out by using a one-tailed binomial hypothesis test with the null hypothesis that 'good SNPs' have an underlying acceptable error rate of ≤5% (H_0: Percent_misclassified ≤ 5%). SNPs with $p < 0.05$ were removed. This filtration method removed 5869 and 24,137 SNPs from the Col/Ler and Col/Cvi SNP lists respectively, after which we regenerated the Ler/Cvi SNP list to create the final lists of 378,743 Col/Ler SNPs, 555,801 Col/Cvi SNPs, and 619,477 Ler/Cvi SNPs (*Supplementary file 2*).

## Data access

Data is deposited under GEO accession number GSE52814 and is also available from the Dryad Digital Repository: http://dx.doi.org/10.5061/dryad.gv536.

## Acknowledgements

We thank Prat Thiru and Bingbing Yuan for assistance with data analysis, Morgan Moeglein for technical assistance, the Whitehead Genome Technology core for strand-specific RNA-seq library preparation, high-throughput sequencing and sharing of equipment and reagents, Hongcang Gu and Alex Meissner for advice on bisulfite sequencing library preparation, members of the Bartel lab for advice and protocols on small RNA library preparation, and Nathan Springer for extensive discussions and comments on the manuscript. RME and CLP are recipients of NSF Graduate Research Fellowships. This work was funded by the NSF (MCB 1121952) and by a Pew Scholars award to MG from The Pew Charitable Trust's Pew Scholars Program in the Biomedical Sciences.

## Additional information

### Funding

| Funder | Grant reference number | Author |
|---|---|---|
| National Science Foundation | MCB 1121952 | Mary Gehring |
| Pew Charitable Trusts | Pew Scholars Program in the Biomedical Sciences | Mary Gehring |
| National Science Foundation | Graduate Research Fellowship | Robert M Erdmann, Colette L Picard |

The funders had no role in study design, data collection and interpretation, or the decision to submit the work for publication.

### Author contributions

DP, MG, Conception and design, Acquisition of data, Analysis and interpretation of data, Drafting or revising the article; RME, ES, Acquisition of data, Analysis and interpretation of data, Drafting or revising the article; CLP, GWB, Analysis and interpretation of data, Drafting or revising the article

## Additional files

### Supplementary files

• Supplementary file 1. Primers used in this study.

• Supplementary file 2. SNPs used in this study.

• Supplementary file 3. Script to classify mRNA-seq reads by strain.

## Major datasets

The following datasets were generated:

| Author(s) | Year | Dataset title | Dataset ID and/or URL | Database, license, and accessibility information |
|---|---|---|---|---|
| Gehring M, Pignatta D, Erdmann RM, Bell GW, Scheer E | 2013 | Data from: Natural epigenetic polymorphisms lead to intraspecific variation in Arabidopsis gene imprinting | https://datadryad.org/resource/doi:10.5061/dryad.gv536 | Available at Dryad Digital Repository under a CC0 Public Domain Dedication. |
| Mary Gehring | 2013 | Data from: Natural epigenetic polymorphisms lead to intraspecific variation in Arabidopsis gene imprinting | http://www.ncbi.nlm.nih.gov/geo/query/acc.cgi?token=glkjwaugdhatvut&acc=GSE52814 | Publicly available at NCBI Gene Exression Omnibus. |

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
