## [Decision Letter]

Thank you for resubmitting your work entitled “Natural epigenetic polymorphisms lead to intraspecific variation in Arabidopsis gene imprinting” for further consideration at *eLife*. Your revised article has been reviewed by Detlef Weigel (Senior editor) and two other peer reviewers. The manuscript has been improved but there are some remaining issues that need to be addressed before acceptance, as outlined below:

You use an impressive dataset of gene expression, DNA methylation and small RNAs from seeds of reciprocal hybrid plants between two Arabidopsis ecotypes to determine how many genes are imprinted in Arabidopsis. Although there have been several reports published in the past two years regarding epigenetic differences within and among Arabidopsis inbred lines, these data go one step farther by showing how these epigenetic differences can be associated with allele-specific imprinting.

While the reviewers would have liked to see a connection between imprinting polymorphisms and phenotypic effects, there was agreement that this is a difficult task. Nevertheless, there were two principal suggestions how the manuscript could be improved:

Despite their impressive dataset, you identify only six examples for allele-specific imprinting, only three of which are investigated in detail. You either need to investigate the other three, or if you tried to investigate them, explain what you found. If the global analysis had a 50% false discovery rate, then this would be important to know.

As importantly, the Discussion of the importance of epigenetic variation being linked or unlinked to genetic variation tends to obscure the important new points in the current work. The Discussion should focus on methylation variation being causal for imprinting differences, regardless of the proximate cause for methylation variation. Of course, it is ultimately important as well to partition methylation variation into ’pure’ methylation variation and methylation variation that is the result of nearby sequence changes, but this need not be the take home message of the current work.

*Reviewer #2* major concerns:

1) From the start, this paper reads like a thesis and not a research paper. So much information is thrown at the reader that I didn't know what is important. For example, the Introduction is 4 pages long and attempts to review this entire field. Entire paragraphs should be removed so the reader doesn't get bogged down. In the Discussion, the writing is very repetitive between the first and last paragraphs on the page. Because of the way it is written (starting with very large datasets and narrowing down to 2-3 confirmed genes) the manuscript seems as if it has a very high data: conclusions ratio.

2) Throughout the manuscript the authors use CHH context methylation as a signature of RNA-directed DNA methylation. In the past year we have seen several high-profile publications demonstrating that CHH methylation can be maintained by CMT2. Therefore, CHH methylation is not just a signature of RdDM. The author's analysis of CHH methylation is out of date and their calculation of the rate of RdDM based on CHH methylation is fundamentally flawed. In the Discussion, the fact that CHH is retained is provided as evidence of functional RdDM. This is simply incorrect logic.

3) The major experimental issue with this manuscript is that some of the genomes that are being analyzed by deep sequencing datasets have not been de novo assembled. Therefore, there is no guarantee that the gene-neighboring TE is even in the same location in the different genome. The bench work in the later figures proves for some examples that the TEs are in the correct locations, but these are just 2-3 out of hundreds analyzed. For example, the authors find 6 examples of allele-specific imprinted genes. However only 3 were confirmed by bench work and resequencing. Did the other 3 simply not have TEs next to those genes?

4) In the small RNA analysis, all sized small RNAs are lumped together, while only 24nt siRNAs direct DNA methylation. Why lump them all together? Instead, the analysis should focus on just the 24mers.

5) Do mutations in the 2-3 confirmed genes, such as HDG3, have seed phenotypes? For this manuscript to be on the Nature/Science/Cell level, the authors would need to show that these genes matter for phenotype and have an imprinted function, and it isn't just transcriptional noise or a random one-off. When studying transposable elements, you can always find one, two or a few out of tens of thousands have some odd expression or regulation pattern. However, this doesn't mean that these are larger modes of regulation, but rather just one or two positions that act strangely. Can the authors provide any evidence that this is a general mechanism and not just something weird that 2-3 out of thousands of genes or transposable elements do?

6) Lastly, as a major comment, I feel that the data visualization does not provide the reader with any way observe the conclusions that the data is showing. For example, from Figure 2 the authors argue that 12 genes have allele-specific imprinting... However, when I look closely at Figure 2 don't see any pattern or even understand what each row is. Sure, this is colorful, but I couldn't see any patterns or observe anything in the data for much of Figure 2, Figure 4, all of Figure 5, and a significant portion of Figure 7. I personally think that the authors should step back, determine what they are trying to show, and then find alternate ways to display their large datasets in a way that the reader can see the same thing.

*Reviewer #3* major concerns:

The authors point out that their ultimate goal is to connect imprinting polymorphisms to seed phenotypes. That is a difficult task, but that connection is not made in this manuscript. The Abstract includes the prediction that natural epialleles are likely to exert their strongest effect on plant development during the reproductive phase. This claim is intriguing, but drawing this conclusion from the data presented is difficult as the imprinting variation noted has not been connected to developmental phenotypes.

Short of making that connection, the results could be extended by investigating in more detail other allele-specific imprinted loci involving transposable elements (currently 3 of 6 are studied in more detail). In addition, the strength of the prediction about variable imprinting behavior could be explored by looking at more examples of intraspecific crosses with hypomethylated loci corresponding to PEGs in other interstrain crosses.

The question of whether the differences in DNA methylation are caused by genetic polymorphisms or not is addressed briefly in the text and Figure 6—figure supplement 3. This important point needs more consideration in the Discussion.

[Editors’ note: a previous version of this study was rejected after peer review, but the authors submitted for reconsideration. The previous decision letter after peer review is shown below.]

Thank you for choosing to send your work, “Natural epigenetic polymorphisms lead to intraspecific variation in Arabidopsis gene imprinting”, for consideration at *eLife*. Your submission has been reviewed by Detlef Weigel in consultation with two outside peer reviewers. Although the work is of interest, we regret to inform you that the findings at this stage are too preliminary for further consideration at *eLife.*

The major concern was that it is difficult to draw definitive conclusions because of the design of the crosses, with an accession that stands out for its low overall methylation level playing a central role, together with the limitations imposed by the relatively small amount of BS-seq data. A further concern was that you did not take advantage of the opportunity to study a connection to a biological phenotype, large seeds when Cvi is the maternal parent.

Reviewer 1:

Substantive concerns:

The authors directly compared expression pattern and DNA methylation in six combinations of inter-accession crosses. They found intraspecific variation in imprinting associated with variation in DNA methylation. The most interesting observation is, however, that the Cvi accession is unique for its big seed phenotype, which is seen when Cvi is the female parent. Cvi is also unique in the low CpG methylation level in gene bodies. The Cvi-specific seed phenotype might be due to Cvi-specific variation in epigenetic state of imprinted gene(s). This very interesting possibility should be proven by identifying the gene(s) responsible for the seed phenotype. If substantiated through the action of specific gene(s), this would greatly elevate the work.

Reviewer 2:

Substantive concerns:

Pignatta and colleagues analyzed gene imprinting in crosses of three different accessions of *A. thaliana* (Col-0, Ler and Cvi) and its link to changes in TE and small RNA composition as well as DNA methylation. These specific accessions were chosen because of their differences in seed size and their degree of genetic divergence.

The authors identified 420 maternally and 159 paternally expressed genes. DNA methylation analysis revealed differentially methylated regions (DMRs) that were in most parts either specific for comparisons between accessions or for comparisons between endosperm and embryo. The authors identified three imprinted genes that were associated with DMRs (endosperm-embryo as well as across-accession comparisons). These genes were P/MEGs depending on the epigenetic configuration of the parents and are thus examples of epialle-specific imprinted genes.

The authors address the interesting question if and how variations in the epigenetic configuration of the parents influence seed and embryo development through variations in imprinting. The authors have chosen an adequate model to address this question and have employed a broad spectrum of analyses. Unfortunately, the study suffers from weaknesses with regard to the technical and statistical procedures; furthermore the conclusions drawn by the authors are not entirely supported by the experimental evidence. In my view, with only three examples of epialleles influencing imprinting in the endosperm/embryo, the validity of extrapolating the results is not provided. The authors state that:

“Given the epigenetic diversity present within the species, we extrapolate that allele-specific imprinting will be widespread.” Cvi is the most highly differentially methylated *A. thaliana* accession analyzed to date (44). If only three epialleles with effect on imprinting could be found in a comparison with this outlier strain, it is questionable that the effect is actually widespread.

On the technical side, I mainly criticize the low coverage of the BS-seq experiments, which to my opinion do not provide enough information for the performed analyses; and the RNA-seq experimental setup with essentially no replication.

The design of the mRNA-seq experiment is questionable. While two replicates were used for the Col-Cvi reciprocal crosses, there is only one for the respective Ler-Cvi crosses. For Col-Ler, the authors combined newly generated data with a previously published dataset (13). Not only does this mean that these crosses were done at completely different time points, the experiment also lacks the necessary replicates. In addition, the authors combined the reads of different replicates before doing the analysis. I recommend that at least three biological replicates should be used to gain enough statistical power for the confident detection of allele-specific gene expression. On a related note, I also wonder why the authors did not use long reads to increase their chances of allele-specific mapping of the reads?

The coverage obtained by BS-sequencing is very low! The Col-0 endosperm sample has 1,7 million reads, which results in a theoretical coverage of 0.5 x genome-wide. There are less than 1 million cytosines sufficiently covered in these samples and I doubt that this allows a representative and unbiased representation of the genome-wide DNA methylation.

---

## [Author Response]

We thank the reviewers for their continued consideration of the manuscript and their comments. In the revision we have further validated additional genes that were identified as allele-specific imprinted genes from the global RNA-seq analysis, including the three genes reference above that were not investigated earlier. We performed RT-PCR on independently isolated endosperm samples (ColxLer, LerxCol, ColxCvi, CvixCol) and sequenced the cloned PCR products by Sanger sequencing (AT4G13460, AT1G26620, AT5G60760), or performed CAPs digestion (AT4G00750), to distinguish the maternal and paternal allelic contributions to gene expression. In agreement with the RNA-seq data, AT4G00750 was a MEG except when Ler was the male in the cross (Figure 1—figure supplement 2). The data for AT5G60760 were also in complete agreement with the RNA-seq data (PEG except when Ler is male in the cross). For AT1G26620 (PEG except when Cvi is the male) the Col-Cvi reciprocal crosses and Col x Ler data agreed with the RNA-seq data, but for Ler x Col there was more maternal allele expression than in the RNA-seq data. For AT4G13460 (PEG except when Ler is the male) Col-Ler reciprocal crosses and the Cvi x Col cross agreed with the RNA-seq data; in the Col x Cvi cross there was greater maternal allele expression. These data are in [Supplementary-material SD5-data] and, combined with our previous validation efforts, further support our global mRNA-seq analysis.

Additionally, we cloned and sequenced the 5’ regions in Ler and Cvi for the three additional allele-specific imprinted genes (AT4G13460, AT1G26620, AT5G60760) where the overlapping embryo-endosperm and strain DMRs fell within a TE. As expected given our mapping and analysis strategy, the methylation differences we observed are not due to absence or major sequence alteration of the TE. The sequencing confirmed the presence of the relevant TE in Ler and Cvi at the same genomic location as in Col. These results are stated in the text and further detailed in the response to reviewer 2 below.

As suggested, we have modulated the Results and Discussion to emphasize methylation variation as causal for the imprinting differences, rather than focusing on whether or not this variation is purely epigenetic.

Finally, as detailed in the response to specific reviewer comments below, we have further edited the manuscript for length, included analysis specifically of 24 nt small RNAs, and revised figures for clarity.

Reviewer #2 major concerns:

*1) From the start, this paper reads like a thesis and not a research paper. So much information is thrown at the reader that I didn't know what is important. For example, the Introduction is 4 pages long and attempts to review this entire field. Entire paragraphs should be removed so the reader doesn't get bogged down. In the Discussion, the writing is very repetitive between the first and last paragraphs on the page. Because of the way it is written (starting with very large datasets and narrowing down to 2-3 confirmed genes) the manuscript seems as if it has a very high data: conclusions ratio*.

The manuscript has been edited for conciseness and clarity, and the Introduction is considerably shorter. With regard to the ratio of data to conclusions, we note that it would have been impossible to identify allele-specific imprinted genes without performing global mRNA-seq experiments from multiple accessions. We have primarily focused the paper on allele-specific imprinting in order to tell a comprehensive story. However, our work has several other novel and important conclusions, some of which we list here. This is the most comprehensive analysis of imprinting in Arabidopsis, and we find that most imprinted genes are consistently parentally biased among strains (Figure 1 and associated supplemental figures); this is an important conclusion in its own right. We present the first data on allele-specific small RNAs from Arabidopsis seeds and find that paternal 24 nt small RNAs are present, despite a widely accepted view in the field (based on small RNA data from whole siliques) to the contrary ([Supplementary-material SD9-data]; Figure 5—figure supplement 3). We find that MEGs are enriched for small RNAs derived from the paternally inherited allele, suggesting that the paternal allele might be specifically silenced by *cis* acting small RNAs (Figure 5—figure supplement 3). We also show that the Cvi strain is strikingly hypomethylated in gene bodies (Figure 3 and associated supplemental figures). Although the functional consequence of this, if any, is presently unknown, this observation is another important contribution to the field.

*2) Throughout the manuscript the authors use CHH context methylation as a signature of RNA-directed DNA methylation. In the past year we have seen several high-profile publications demonstrating that CHH methylation can be maintained by CMT2. Therefore, CHH methylation is not just a signature of RdDM. The author's analysis of CHH methylation is out of date and their calculation of the rate of RdDM based on CHH methylation is fundamentally flawed. In the Discussion, the fact that CHH is retained is provided as evidence of functional RdDM. This is simply incorrect logic*.

We agree with the reviewer that the genome-wide % CHH methylation is not strictly informative of RdDM, but disagree that our analysis is fundamentally flawed. As shown by Zemach et al (*Cell* 153, 2013), CMT2 is primarily responsible for DNA methylation in heterochromatic TEs. The TEs demethylated in endosperm are not of the CMT2-type; they are primarily euchromatic TEs of the Helitron class (Figure 4—figure supplement 4).

*3) The major experimental issue with this manuscript is that some of the genomes that are being analyzed by deep sequencing datasets have not been de novo assembled. Therefore, there is no guarantee that the gene-neighboring TE is even in the same location in the different genome. The bench work in the later figures proves for some examples that the TEs are in the correct locations, but these are just 2-3 out of hundreds analyzed. For example, the authors find 6 examples of allele-specific imprinted genes. However only 3 were confirmed by bench work and resequencing*. *Did the other 3 simply not have TEs next to those genes?*

We have resequenced the 5’ regions containing the DMR and TE for the three other genes, AT4G13460, AT5G60760 and AT1G26620, from Ler and Cvi. In all cases the annotated TE is present at the same genomic location as in Col and contains a few SNPs but no major sequence alterations. Specifically, for AT1G26620 we sequenced 3400 bp 5’ of gene, encompassing the DMR in AT1TE29660. The TE (at -900 to -1800) was present in the same location in Ler and Cvi as in Col, although in Cvi there was a 183 bp insertion at -420 and a 321 bp deletion at -2870 bp. These sequences differences are outside of the TE. For AT4G13460 we sequenced the entire 5’ region (2080 bp). Apart from a few SNPs there were no DNA sequence changes in Cvi. In Ler there were two small deletions (45 bp and 38 bp) 329 and 684 bp 5’ of the relevant TE, but no changes to the TE itself. For AT5G60760 we sequenced 2700 bp 5’ of the gene, encompassing AT5TE87930 and AT5TE87925, two contiguous TEs in the 5’ region. Both TEs are unaltered in Ler and Cvi except for a few SNPs.

*4) In the small RNA analysis, all sized small RNAs are lumped together, while only 24nt siRNAs direct DNA methylation. Why lump them all together? Instead, the analysis should focus on just the 24mers*.

RdDM is primarily directed by 24 nt small RNAs (the Pol IV-RdDM pathway) but it has also been shown that RDR6-dependent 21 nt small RNAs can direct DNA methylation of TEs (see for example Pontier et al, *Mol Cell*, 2012 and Nuthikattu et al, *Plant Physiol*, 2013). However, the majority of small RNAs overlapping DMRs are of the 24 nt type. In response to the reviewer, we present the analysis on 24mers only in Figure 5, Figure 5—figure supplement 2, and Figure 5—figure supplement 3. This revision does not alter our conclusions.

*5) Do mutations in the 2-3 confirmed genes, such as HDG3, have seed phenotypes? For this manuscript to be on the Nature/Science/Cell level, the authors would need to show that these genes matter for phenotype and have an imprinted function, and it isn't just transcriptional noise or a random one-off. When studying transposable elements, you can always find one, two or a few out of tens of thousands have some odd expression or regulation pattern. However, this doesn't mean that these are larger modes of regulation, but rather just one or two positions that act strangely*. *Can the authors provide any evidence that this is a general mechanism and not just something weird that 2-3 out of thousands of genes or transposable elements do?*

We addressed how widespread allele-specific imprinting might be by looking at variation for all DMRs within the population (Figure 7). Based on this analysis, we believe that there are more examples of allele-specific imprinting than those that we specifically discovered in our Col/Ler/Cvi datasets, and we demonstrated this in additional strains for AT3G14205 and AT2G32370 (Figure 6, Figure 6—figure supplement 2). From our analysis we projected that maximally 11% of imprinted genes would be subject to allele specific imprinting. We would have been surprised if allele-specific imprinting was rampant. The ’general mechanism’ here is that natural variation in the methylation status of genes can alter whether or not they are imprinted. Even if a TE losing (or gaining) methylation in a particular strain is an infrequent event, the point is that these changes in methylation have consequences for gene expression. The analysis of population variation in DNA methylation shows that 15% of DMRs (the strongly bimodal class; Figure 7) exist in highly methylated and lowly methylated states within the species. While not all of these methylation differences will impact imprinting, we have shown that it does for AT2G32370 and AT3G14205 (Figure 6 and Figure 6—figure supplement 2). We do not yet know if *HDG3* (AT2G32370) or other allele-specific imprinted genes have phenotypes when mutated.

*6) Lastly, as a major comment, I feel that the data visualization does not provide the reader with any way observe the conclusions that the data is showing. For example, from*
Figure 2
*the authors argue that 12 genes have allele-specific imprinting... However, when I look closely at*
Figure 2
*don't see any pattern or even understand what each row is. Sure, this is colorful, but I couldn't see any patterns or observe anything in the data for much of*
Figure 2*,*
Figure 4*, all of*
Figure 5*, and a significant portion of*
Figure 7*. I personally think that the authors should step back, determine what they are trying to show, and then find alternate ways to display their large datasets in a way that the reader can see the same thing*.

We have amended the figures as follows: part A has been removed from Figure 2, the heat map showing methylation difference at imprinted genes has been replaced with a methylation metagene analysis in Figure 5, and Figure 7 has been modified to include examples of each class of DMR and part C has been modified for visual consistency with part A.

Reviewer #3 major concerns:

*The authors point out that their ultimate goal is to connect imprinting polymorphisms to seed phenotypes. That is a difficult task, but that connection is not made in this manuscript. The Abstract includes the prediction that natural epialleles are likely to exert their strongest effect on plant development during the reproductive phase. This claim is intriguing, but drawing this conclusion from the data presented is difficult as the imprinting variation noted has not been connected to developmental phenotypes*.

We have tried to clarify in the Abstract and text that what we have shown is an expression phenotype. We are careful not to claim a developmental phenotype beyond speculating that this is a possible outcome of the expression phenotype we observed.

*Short of making that connection, the results could be extended by investigating in more detail other allele-specific imprinted loci involving transposable elements (currently 3 of 6 are studied in more detail). In addition, the strength of the prediction about variable imprinting behavior could be explored by looking at more examples of intraspecific crosses with hypomethylated loci corresponding to PEGs in other interstrain crosses*.

As detailed above, in the revised version we have further validated allele-specific imprinting for four genes and demonstrated that the all of the TEs associated with allele-specific imprinted genes are present at the same genomic location in all three strains.

*The question of whether the differences in DNA methylation are caused by genetic polymorphisms or not is addressed briefly in the text and*
Figure 6—figure supplement 3*. This important point needs more consideration in the Discussion*.

As suggested in the summary review statement above, we have focused on methylation variation being casual for differences in imprinting rather than focusing on whether or not this variation is purely epigenetic or whether it has a genetic basis.

[Editors’ note: the author responses to the previous round of peer review follow.]

We thank the reviewers for their careful review of the original submission. We have significantly strengthened and extended the manuscripts’ Conclusions by addressing the reviewers’ comments. The following major changes have been made; we have estimated how widespread allele-specific imprinting might be by analyzing methylation variation at the population scale for regions targeted for endosperm demethylation (Figure 7), predicted and confirmed an additional example of allele-specific imprinting based on this analysis (Figure 6—figure supplement 2), performed additional analyses on the allele-specific imprinted gene *HDG3* (Figure 6), and performed three endosperm RNA-seq replicates for each set of reciprocal crosses (Figure 1, [Supplementary-material SD3-data]) . The Results section has been reorganized for greater clarity and the Introduction and Abstract revised to better reflect the aims and Conclusions of our study.

Detailed responses to each reviewer’s comments follow.

Reviewer 1:

*Substantive concerns*:

*The authors directly compared expression pattern and DNA methylation in six combinations of inter-accession crosses. They found intraspecific variation in imprinting associated with variation in DNA methylation. The most interesting observation is, however, that the Cvi accession is unique for its big seed phenotype, which is seen when Cvi is the female parent. Cvi is also unique in the low CpG methylation level in gene bodies. The Cvi-specific seed phenotype might be due to Cvi-specific variation in epigenetic state of imprinted gene(s). This very interesting possibility should be proven by identifying the gene(s) responsible for the seed phenotype. If substantiated through the action of specific gene(s), this would greatly elevate the work*.

We have uncovered interesting candidate genes that have the potential to contribute to seed size, including genes that are differentially expressed when Cvi is the male or female parent and genes, such as *HDG3*, that are differentially imprinted between Cvi and other strains. We have not made an explicit analysis of their connection to seed phenotype for two reasons:

1) Alonso-Blanco et al (PNAS, 1999) performed QTL mapping of seed size differences between Ler and Cvi and discovered 11 significant loci. The fact that seed size is not a single locus trait means that uncovering the network of genes that contribute to differences, regardless of whether or not they are imprinted, will be a considerable undertaking and thus the subject of a different study.

2) The primary aim of our study was not to determine what genes are responsible for making Cvi seeds larger than Col or Ler seeds, but to understand whether or not variation existed in the population for gene imprinting, and if it did whether this might be due to epigenetic differences between strains. We believe we have demonstrated that this variation exists.

Reviewer 2:

*Substantive concerns*:

*Pignatta and colleagues analyzed gene imprinting in crosses of three different accessions of* A. thaliana *(Col-0, Ler and Cvi) and its link to changes in TE and small RNA composition as well as DNA methylation. These specific accessions were chosen because of their differences in seed size and their degree of genetic divergence*.

*The authors identified 420 maternally and 159 paternally expressed genes. DNA methylation analysis revealed differentially methylated regions (DMRs) that were in most parts either specific for comparisons between accessions or for comparisons between endosperm and embryo. The authors identified three imprinted genes that were associated with DMRs (endosperm-embryo as well as across-accession comparisons). These genes were P/MEGs depending on the epigenetic configuration of the parents and are thus examples of epialle-specific imprinted genes*.

*The authors address the interesting question if and how variations in the epigenetic configuration of the parents influence seed and embryo development through variations in imprinting. The authors have chosen an adequate model to address this question and have employed a broad spectrum of analyses. Unfortunately, the study suffers from weaknesses with regard to the technical and statistical procedures; furthermore the conclusions drawn by the authors are not entirely supported by the experimental evidence. In my view, with only three examples of epialleles influencing imprinting in the endosperm/embryo, the validity of extrapolating the results is not provided. The authors state that*:

*“Given the epigenetic diversity present within the species, we extrapolate that allele-specific imprinting will be widespread.” Cvi is the most highly differentially methylated* A. thaliana *accession analyzed to date (*[44]*). If only three epialleles with effect on imprinting could be found in a comparison with this outlier strain, it is questionable that the effect is actually widespread*.

We disagree that Cvi is an outlier or anomalous strain with regard to imprinting and thus the source of the maximal number of allele- specific imprinted genes. First, our data show that of the 12 allele-specific imprinted genes we identified, 5 are variable in Cvi and 7 are variable in Ler ([Supplementary-material SD6-data]). 4 of the Cvi genes and 6 of the Ler genes are associated with an embryo-endosperm DMR that is variably methylated among the Col, Ler, and Cvi strains. We further focus on three genes, AT2G32370 (not imprinted in Cvi), AT3G14205 (not imprinted in Cvi), and AT2G34890 (not imprinted in Ler). Thus the phenomenon of allele-specific imprinting is not restricted to Cvi.

Second, to convince the reviewer of our assertion, in the revision we directly addressed how widespread allelic variation in imprinting might be. To this end we took the set of embryo-endosperm CG DMRs we found in any of our comparisons (the union) and queried these regions against the Ecker whole genome BS-seq data from 140 strains for which vegetative methylation patterns are known (Schmitz et al, Nature, 2013). We determined how variably methylated these regions were among all strains with sufficient BS-seq data (Figure 7). Many of the DMRs (41%) show little variability in DNA methylation within the population – we define these as very low range DMRs (less than 0.2 mean methylation difference across all strains) and low range (between 0.2 and 0.4 mean methylation difference across all strains). For example, the endosperm DMR at *FWA* falls into this category; it is highly methylated in all strains.

Imprinted genes associated with invariable DMRs are likely to be targeted for demethylation, and imprinted, in all strains. However, we also find bimodal DMRs, with clustering of mean methylation towards one end of the spectrum, but with one or a few clear outliers. We define these as strong (n=1654) and weak (n=1545). Of the shared endosperm and strain CG DMRs associated with 8 allele-specific imprinted genes, 6 are in the strong or weak bimodal category, including those at AT2G32370 and AT3G14205 ([Supplementary-material SD6-data] and Figure 7 ). We further show that of the union set of imprinted genes we identified in this study (n=388), 27 MEGs and 17 PEGs, including 4 PEGs we find as exhibiting allele-specific imprinting among Col, Ler, and Cvi, are associated with DMRs that are strongly bimodal for methylation within the population. This would represent the maximal number of allele-specific imprinted genes, approximately 11% of all imprinted genes. Our genome-wide analysis identified another strain, in addition to Cvi, where the DMR at AT3G14205 was hypomethylated. We showed that reciprocal crosses with this strain also exhibit allele-specific imprinting (Figure 6—figure supplement 2).

*On the technical side, I mainly criticize the low coverage of the BS-seq experiments, which to my opinion do not provide enough information for the performed analyses; and the RNA-seq experimental setup with essentially no replication*.

To identify embryo-endosperm differentially methylated regions (DMRs), we only compared regions of the genome where we had sufficient coverage. In the revised version we show that the DMRs we identified were consistent with previous studies ([11]; [23], comparison in Figure 4—figure supplement 5 ). In the revised version, we used our analysis method to identify DMRs from the [23] Col x Ler and Ler x Col embryo and endosperm BS-seq data (from tissue isolated at 7-8 DAP). These DMRs are now included in the analyses of the relationship between DNA demethylation and imprinting at several points ([Supplementary-material SD6-data], Figure 4—figure supplement 2, and Figure 7).

We agree that our endosperm BS coverage is low. Our intent was not to identify all differentially methylated regions between embryo and endosperm. Rather, we wanted to know whether regions targeted for demethylation in the endosperm were ever variably methylated among strains, which we have shown (to identify strain-DMRs between Col, Ler, and Cvi we used the embryo data, where our coverage was much better). Despite our repeated efforts at optimization and considerable sequencing, the same starting amount of endosperm gDNA did not make a BS-seq library as diverse as embryo gDNA collected and treated at the exact same time; most endosperm BS-seq reads were discarded because they were redundant (mapped to same start position).

*The design of the mRNA-seq experiment is questionable. While two replicates were used for the Col-Cvi reciprocal crosses, there is only one for the respective Ler-Cvi crosses. For Col-Ler, the authors combined newly generated data with a previously published dataset (*[13]*). Not only does this mean that these crosses were done at completely different time points, the experiment also lacks the necessary replicates*. *In addition, the authors combined the reads of different replicates before doing the analysis. I recommend that at least three biological replicates should be used to gain enough statistical power for the confident detection of allele-specific gene expression. On a related note, I also wonder why the authors did not use long reads to increase their chances of allele-specific mapping of the reads?*

As suggested by the reviewers, we have now included three endosperm biological replicates for each of the crosses used to identify imprinted genes. The additional RNA-seq libraries were sequenced using 80 bp reads. We no longer combine read counts or include our 2011 data. To call a gene imprinted in a particular set of reciprocal crosses (Col -Ler, Col- Cvi, or Ler- Cvi) it had to meet all of our criteria for imprinting in at least two of the three replicate sets. We chose to use two of three rather than three of three because of variability in the sequencing depth of our libraries. In the revised version we have also validated the mRNA-seq data by using the miSeq platform; we isolated 2 additional endosperm biological replicates for each set of reciprocal crosses and one additional embryo and performed RT -PCR for 25 genes from these sets and sequenced the PCR amplicons. The results agreed quite well with the mRNA-seq data (see [Supplementary-material SD5-data]).

*The coverage obtained by BS-sequencing is very low! The Col-0 endosperm sample has 1,7 million reads, which results in a theoretical coverage of 0.5 x genome-wide. There are less than 1 million cytosines sufficiently covered in these samples and I doubt that this allows a representative and unbiased representation of the genome-wide DNA methylation*.

We agree that the Col-0 endosperm coverage is very low; therefore it also contributed few reads to the overall analysis compared to the many more reads derived from other libraries. It was never used as a stand-alone dataset to draw broad conclusions on genome-wide methylation or DMRs.